

# On the efficacy of downscaled GRACE total water storage products

Maya Raghunath Suryawanshi[1], K Satish Kumar[1], Amin Shakya[2], Shard Chander[3], Bhaskar R. Nikam[4], Nagesh Kumar Dasika[5], Bramha Dutt Vishwakarma[1,6]

[1]Interdisciplinary Centre for Water Research, Indian Institute of Science, Bengaluru, 560012, India
[2]Faculty of Geo-Information Science and Earth Observation, University of Twente, Enschede, 7500 AE, The Netherlands
[3]Land Hydrology Division, Space Applications Centre, Indian Space Research Organisation, Ahmedabad, 380015, India
[4]Indian Institute of Remote Sensing, Indian Space Research Organisation, Dehradun, 248001, India
[5]Department of Civil Engineering, Indian Institute of Science, Bengaluru, 560012, India
[6]Centre for Earth Sciences, Indian Institute of Science, Bengaluru, 560012, India

*Correspondence to*: Maya Raghunath Suryawanshi (maya2509.surya@gmail.com)

**Abstract.** The Gravity Recovery and Climate Experiment (GRACE) satellite mission provides estimates of total water storage anomalies (TWSA) in terms of "mascons" (mass concentration blocks), representing integrated changes in surface

water, soil moisture, and groundwater. However, the coarse spatial resolution of GRACE mascons (~3°) limits its utility for regional-scale hydrological analysis. Although several downscaling methods have been proposed to improve resolution, none have been comprehensively validated against in-situ observations. In this study, we are validating both native and downscaled GRACE TWSA products using well-based in-situ groundwater observations across India. Furthermore, we develop an improved downscaling method by modifying the approach of Vishwakarma et al. (2021), i.e., incorporating mass

conservation constraints at the native GRACE resolution instead of catchment scale (Vishwakarma et al., 2021) to better capture spatial variability at a finer 0.5° grid scale. We assessed the efficacy of our downscaled product and two existing downscaled GRACE products (Vishwakarma et al., 2021; Gao and Soja, 2024) using performance metrics such as gain in correlation (r) and gain in root-mean-square error (RMSE) relative to native GRACE product. The entire India is covered by 50 mascons. Out of these 50 mascons groundwater observations are available only at 22 mascons. Our modified approach

shows improved performance across 12 mascons in India, with median gains in r ranging from 0.58% to 8.84% and gain in RMSE improvements from 0.24% to 23.59%. Whereas, the Vishwakarma et al. (2021) and Gao and Soja (2024) methods perform well in only 3 and 10 mascons, respectively. These results demonstrate that our enhanced downscaling approach provides more accurate and spatially resolved estimates of groundwater storage changes, offering a valuable tool for regional water resource assessments in India. Further, the downscaled GRACE product developed in this study (Mascon wise Mass

Conservation product) is publicly accessible at https://figshare.com/articles/dataset/GRACE_downscaled_TWSA_product_using_Mascon_wise_Mass_Conservation/29196 617.



## 1 Introduction

Freshwater is a critical resource for sustainable development, therefore, understanding changes in its availability is one of the
top priorities for researchers, governments and water resource managers. The fundamental process which regulates
freshwater availability is the water cycle and by studying its components, we enhance our understanding of water
availability. Total water storage (TWS) is an important part of the water budget, which is written as a sum of soil moisture,
vadose zone moisture, groundwater, surface water, snow water, and canopy water storage. Since TWS provides an estimate
of water available to sustain lean phases, understanding the spatiotemporal variations in TWS and their drivers is essential to
manage future water crises.

Since collecting global subsurface in-situ data is nearly impossible and traditional remote-sensing satellites only provide
surface information, TWS was not observable until the launch of the Gravity Recovery and Climate Experiment (GRACE)
satellite mission in 2002.  GRACE measures precise changes in the distance between two identical satellites flying in the
same orbit, one after the other. These range measurements are then used to estimate the gravitation potential variations from
one month to the other. Since gravity field perturbations at monthly scale are driven by hydrology, potential anomalies are
converted to TWS anomalies (TWSA) (Tapley et al. 2004; Wahr et al. 2004). The signal is a superposition of signals from
all TWS compartments mentioned earlier and has advanced hydrology significantly in the last two decades.

TWSA from GRACE is difficult to validate due to unavailability of in-situ TWSA data (Scanlon et al., 2016) but not
impossible if one compartment of TWSA is changing rapidly while others stay table, especially over India where
groundwater decline is alarming (Sarkar et al., 2020).  Another major challenging task to deal with is a coarse spatial
resolution (~3°x3°) of GRACE data. Several hydrological, and agricultural studies demand regional/local scale inputs.
Already several attempts to downscale GRACE TWSA have been realized (Vishwakarma et al., 2021; Zhang et al., 2021;
Schumacher et al., 2018; Tian et al., 2017). However, advances in data techniques and constantly improving data products,
provide scope for improvements. Broadly downscaling methods are categorised in two types: Model-based/Dynamic and
data-based/Statistical method. The model-based approach is physically based, strongly depends on boundary conditions and
computationally expensive. Whereas in the data-based approach an empirical relationship is developed between coarse scale
variables and fine scale variables. Owing to its cheap computation and easy implementation, model-based approach gained
popularity amongst researchers. In traditional data-based downscaling approach 1. GRACE data serve as truth 2. Dependent
high-resolution variables are averaged to coarse resolution of GRACE 3. Model is trained at low resolution to obtain
coefficients (link between the variables) 4. Then fine scale data with coefficients is used to compute downscaled product.
Indeed, this approach rests on the hypothesis that the physical and hydrological processes linking fine and coarse scale
variables are identical at all resolution (i.e., same coefficient for all finer grids lying within a coarse grid of GRACE data).
However, it suffers from two major drawbacks, first, GRACE data does contain some uncertainties. Thus, it cannot be used
as the only reference. Second drawback lies in its hypothesis i.e., processes linking fine and coarse scale variables are



65 identical at all resolutions. This cannot be true owing to differences in the spatial variability of various processes which demands variable coefficient in space.

Thus, to overcome the above two drawbacks a novel data driven approach with a twist of data assimilation approach is implemented by very few researchers (Vishwakarma et al. 2021 and Gou and Soja 2024). In this modified approach, the model is trained at fine resolution and both GRACE and hydrologic model TWS anomalies (TWSA) are considered as

70 references. Specifically, long term-trend and special variability are preserved in GRACE data and hydrologic model, respectively. Major limitation of both products is that they lack validation. Further, Gou and Soja (2024) study does not regress on residuals of predictor variables suggesting downscaling relies on default trend and seasonality of GRACE data. Although Vishwakarma et al. (2021) worked on residuals they removed the trend and seasonality at catchment scale instead of GRACE native scale (~3°) which is finer.

75 In the present study we aim to resolve the major short coming of previous studies with region restricted to India by building a dataset for validation of GRACE TWSA at fine and at coarse resolution using in situ well observations. Additionally, we generated a new downscaled GRACE product at 0.5° resolution, improving upon the approach of Vishwakarma et al. (2021). Specifically, our downscaling is performed using partial least squares regression (PLR), incorporating mass conservation at a finer spatial scale of approximately 3° × 3° i.e., mascon resolution, in contrast to the catchment-scale approach used by

80 Vishwakarma et al. (2021). We evaluate the performance of the Jet Propulsion Laboratory (JPL) GRACE mascon (mass concentration block) solution RL06M, along with its downscaled versions, using an in situ well observations. For clarity, we refer to our newly developed downscaled product as the Mascon-wise Mass Conserved product (MMC product), the Vishwakarma et al. (2021) product as the Catchment-wise Mass Conserved product (CMC product), and the GRACE downscaled product developed by Gou and Soja (2024) using a deep learning approach as the Deep Learning product (DL

85 product).

## 2 Study region and data

### 2.1 Study region

India is the seventh-largest country in the world, covering a total area of 297 million hectares, and it ranks first in terms of population. It receives an average annual precipitation of 118 cm mostly concentrated in a few months, which is insufficient

90 to meet the demands of the country's large agriculture-based population through multiple cropping seasons. As a result, groundwater resources have been over-exploited. According to a report by the Central Ground Water Board (CGWB), India withdrew 239.16 billion cubic meters of groundwater in 2022. Notably, northwest India plays a significant role in this depletion, with groundwater levels declining at an alarming rate of 1.5 cm per year, primarily due to intensive irrigation practices (Mishra et al., 2024). However, there are discrepancies in the groundwater withdrawal rates in southern India.

95 Mondal & Lakshmi (2021) reported a positive trend in groundwater levels in southern India, whereas other studies have shown a significant decline (Shah, 2009).





## 2.2 Data

In this study we downscale GRACE TWSA from its native resolution of approximately 3° to 0.5°. One option is to process level 2 spherical harmonic solutions from GRACE data centres, which requires several post processing steps to reach
gridded mass change products. Since the aim is to highlight downscaling approaches, we choose an already existing level 3 product at native resolution. GRACE mascon solutions for the period between 2004 and 2015 provided by JPL (RL06M) are used in the present study (http://grace.jpl.nasa.gov). Mascons over India for the month of January 2004 are shown in Figure 1a where each block is a mascon. Also note that mascon represents total water storage anomaly w.r.t. baseline period of 2004 to 2009 (January-December) in terms of centimetre of equivalent water height (EWH).

### 2.2.1 GRACE terrestrial water storage anomalies

In this study we downscale GRACE TWSA from its native resolution of approximately 3° to 0.5°. One option is to process level 2 spherical harmonic solutions from GRACE data centres, which requires several post processing steps to reach gridded mass change products. Since the aim is to highlight downscaling approaches, we choose an already existing level 3 product at native resolution. GRACE mascon solutions for the period between 2004 and 2015 provided by JPL (RL06M)
are used in the present study (http://grace.jpl.nasa.gov). Mascons over India for the month of January 2004 are shown in Figure 1a where each block is a mascon. Also note that mascon represents total water storage anomaly w.r.t. baseline period of 2004 to 2009 (January-December) in terms of centimetre of equivalent water height (EWH).

### 2.2.2 Central groundwater board groundwater level

The central ground water board (CGWB), India, provides seasonal measurements (January, March/April/May, August,
November) of in situ groundwater levels (GWLs, in terms of depth below ground) across India (https://indiawris.gov.in/wris/#/). Here we used quality controlled GWL measurements over the period between 2004 and 2015 which includes different types of wells such as Dug, Bore, Dug cum Bore, Tube, Slim hole as well as unknown (Figure 1b). To obtain quality controlled measurements only those wells are selected where positive GWL measurements are available for at least three months in a year without repeating any measurement throughout the study period (2004-2015).
The obtained quality controlled GWLs are used to validate the GRACE groundwater storage changes (GWSC).

### 2.2.3 Soil moisture storage and surface water storage

We used climate data store's monthly gridded fractional soil moisture data which is available at 0.25°x0.25° spatial resolution in $m^3/m^3$ (https://cds.climate.copernicus.eu). This is active as well as passive satellites derived soil moisture product capable of observing soil moisture up to a maximum depth of 5 cm. Also, reservoir storage dataset of two largest
reservoirs; Govind Ballabh Pant and Indira Sagar in India are used to study the contribution of surface water storage in





TWSA and impact of its removal on final GRACE derived GWSC validation. Figure 1b represents the inset of these two reservoirs (Govind Ballabh Pant and Indira Sagar) and they lie in mascon 27 and 33 respectively (Figure 1b).

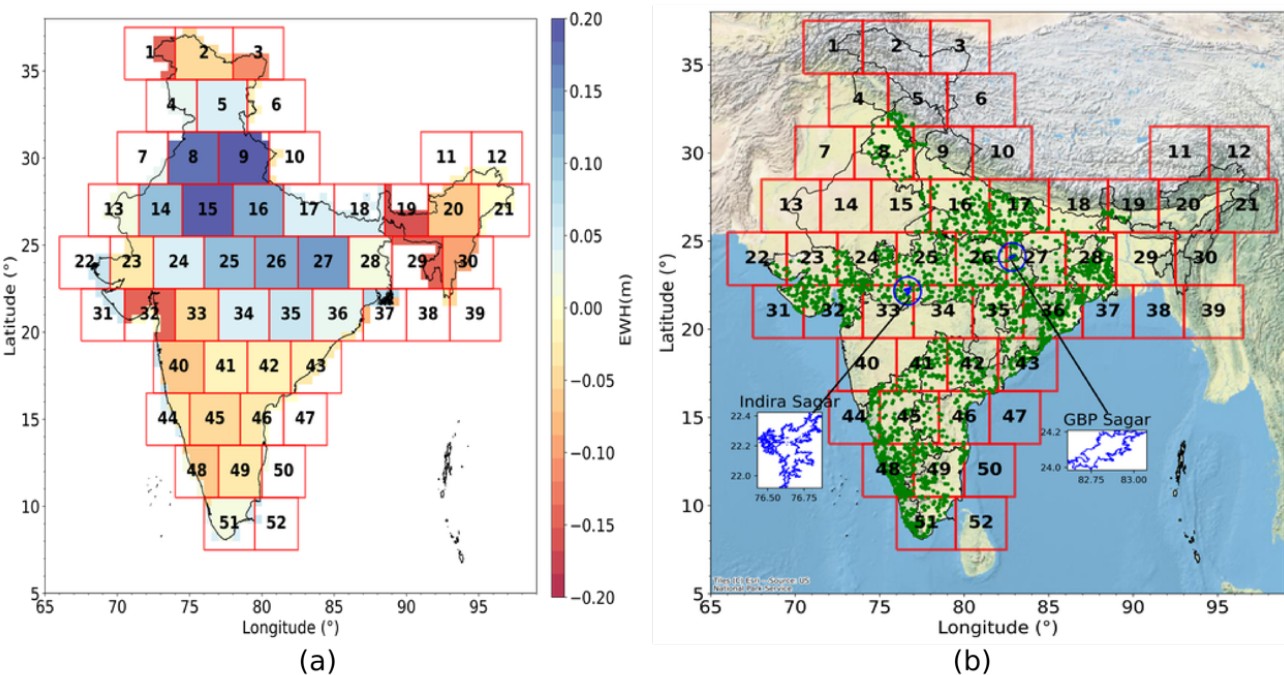

**Figure 1.** (a) TWSA for January 2004 over India, expressed in meters of equivalent water height (EWH), obtained from JPL mascon data. The red boxes indicate the mascon boundaries, and the numbers denote the respective mascon numbers. (b) Green dots represent the filtered groundwater wells across India. The blue-outlined zoomed-in areas show the boundaries of the two large reservoirs in India: Govind Ballabh Pant (GBP) Sagar and Indira Sagar, located in mascon 27 and mascon 33, respectively.

**2.2.4 WaterGAP Global Hydrology Model terrestrial water storage anomalies**

We used monthly TWS output of WaterGAP Global Hydrology Model (WGHM) at spatial resolution of 0.5°x0.5°(https://doi.org/10.1594/PANGAEA.948461)**.** This TWS product comprises of various water stores near the surface of the Earth, such as canopy water storage, snow water equivalent, soil moisture, and so on along with land use information on groundwater aquifers, river streams, lakes, and wetlands. To align with GRACE TWSA base line, we generated TWSA from WGHM by removing mean of baseline period (2004 to 2009).





### 2.2.5 Global Land Assimilation Precipitation, Evapotranspiration and Runoff

We also used monthly precipitation, evapotranspiration and runoff (surface runoff, baseflow-groundwater runoff and snow melt) from NASA Global Land Data Assimilation System Version 2 (GLDASV2.1) available at spatial resolution of 0.25° x 0.25°.(https://disc.gsfc.nasa.gov/datasets/GLDAS_NOAH025_M_2.1/summary?keywords=GLDAS).  All  dataset  used  are

summarized in Table 1.

**Table 1.** Data used in the present study.

| Type of observation | Parameter | Spatial Resolution | Source | Duration |
|---|---|---|---|---|
| **Satellite** | Total Water Storage Anomalies (TWSA) (Equivalent water height, meter) | ~3° | GRACE JPL mascon | January 2004 - December 2015 |
| | Soil Moisture ($m^3/m^3$) | 0.25° | Climate Data Store | |
| **In-situ** | Groundwater level (m) | Point data | Central Ground Water Board (CGWB) | |
| | Reservoir water storage (billion cubic meter (bcm)) | | India- Water Resource Information System (WRIS) | |
| **Hydrologic Model** | Total water storage (Equivalent water height, meter) | 0.5° | WaterGap Global Hydrology Model (WGHM) | |
| | Precipitation ($kg/m^2/s$), Evapotranspiration ($kg/m^2/s$), and Runoff ($kg/m^2$) | 0.25° | Global Land Assimilation (GLDAS) | |
| **Existing GRACE downscaled product** | Total Water Storage Anomalies (TWSA) (Equivalent water height, meter) | 0.5° | Vishwakarma et al., 2021 | |
| | | | Gou and Soja, 2024 | |





## 3 Methodology

### 3.1 Validation

In the present study, GRACE derived ground water changes obtained after removing reservoir water storage and surface soil moisture are validated against in-situ groundwater storage measurements using methodology shown in Figure 2. However, the in-situ GWL measurements, reservoir water storages and soil moisture are in different units. Thus, these quantities are converted to metre (m) equivalent of water height (EWH). The procedure used to bring each of the quantities in EWH unit is as follows:

**3.1.1 Conversion of GWL changes to groundwater storage changes**

To compute groundwater storage changes (GWSC, m EWH), quality controlled GWL change (GWLC) is multiplied with Specific Yield (SY). Where SY is a dimensionless factor that indicates the fraction of total ground water volume that would yield under gravity and is used to convert water level to water storage. SY depends on the distribution of pores, their shape and grain size (Lv et al. 2021).
Fundamentally it is a hydrogeological property of aquifer.

The SY values at quality-controlled wells are extracted from hydrogeology map of India from Bhanja et al. (2016). However, the map is in a standard PNG format image and does not contain any geographic coordinates. Thus, its geo-referencing is done, and a geo-tiff image is generated. Next, SY values in the form of RGB (Red Green Blue) colour combinations (not exactly the aquifer type) are extracted at
respective well locations. Using k-means clustering algorithm k (=8) number of clusters (each representing unique aquifer type) are formed from these multiple RGB combinations. To these k clusters SY values are assigned as mentioned in Table 2. The obtained SY is multiplied with quality controlled GWLC to give reference groundwater storage changes (Ref-GWSC).

**3.1.2 Conversion of reservoir storage from billion cubic meter (bcm) to Equivalent Water Height (m)**

The daily reservoir storages obtained at the reservoirs are converted to monthly storages by averaging. To convert billion cubic meter into m EWH, storges are divided by area of the mascon where the reservoir is located (i.e., storage ($m^3$) / mascon area ($m^2$)).





### 3.1.3 Conversion of fractional soil moisture to soil moisture storage (in EWH (m)),

To convert the fractional soil moisture to soil moisture storage (in EWH (m)), obtained values are
multiplied by 0.05. As it indicates water content only for the maximum top 5 cm (0.05 m) layer of the
soil.

GRACE TWSA are in centimetres of EWH (cm) which are converted to meters of EWH (m).
Thereafter, contemporaneous datasets of four variables i.e., Quality controlled well measurements,
surface water storages, soil moisture and GRACE TWSA are analysed. GRACE provides TWS
anomalies with respect to the mean TWS over 2004 to 2009, whereas other variables are not anomalies
hence they are also converted to anomalies by removing a mean over 2004 to 2009. This can be
implemented for all variables except well observations which are available at quarterly interval.
Therefore, to solve this issue difference with respect to the previous value is calculated as the difference
in anomalies is equal to the difference in the actual values at respective times. This can be understood
from following equations.

$$TWSA_i = TWS_i - LM_{2004-2009} \tag{1}$$

$$TWSC_{ij} = TWSA_i - TWSA_j = TWS_i - TWS_j \tag{2}$$

$$GWSC_{ij} = TWSC_{ij} - SWSC_{ij} - SMSC_{ij} \tag{3}$$

Where $i$ = present month (Jan/May/August/November) and $j$ represents the data for the previous
month. The generated GRACE TWSA differences are referred to as GRACE-GWSC after removal of
surface water storage changes and soil moisture storage changes. These changes are validated against
Ref-GWSC. To validate the GRACE product at its native resolution, averaging of quality controlled
GWLs is done over a mascon whereas to validate the downscaled product, averaging is done over a grid
of 0.5°.

### 3.2 Statistical downscaling to generate mascon wise mass conservation downscaled product

We used partial least square regression (PLR) method to downscale GRACE TWSA to 0.5° from its
native resolution of about 3°. Flow chart of the methodology is as shown in Figure 3. Our method is a
modified version of the method elaborated in Vishwakarma et al. (2021) which uses regression model
as shown in Eq.4.



$S = L \times H$                                                                        (4)

Where S represent predictand matrix with n (total number of months) x g (total number of 0.5°x0.5°

grids in a mascon) dimensions. WGHM TWSA performs the role of this predictand matrix. L is a

predictor matrix i.e. observations matrix. Its dimensions are nxd. Where d columns are of precipitation

(P), evapotranspiration (ET) and runoff (R) and GRACE TWSA. Eq. 1 is solved to obtain H (prediction

matrix, d x g) using Vishwakarma et al. 2021 methodology and put back in Eq. 1 to obtain final

downscaled product. The modification we implemented here is that mass is conserved at mascon scale

instead of catchment scale i.e, for whole set of operations Vishwakarma et la. (2021) unit was a

catchment; in this study it is changed to a mascon.


**Figure 2.** Methodology for validating GRACE mascon at native resolution.





We used high resolution (0.25°x0.25°) product of P, ET, and R from Global Land Assimilation system
(GLDAS) which are resampled to 0.5° by averaging and are converted into EWH (m) unit. But GLDAS
P (variable name is Rain_f_tavg) and ET (Evap_tavg) is measured in kg/m$^2$/s. As each monthly average
quantity has units of per 3 hours to convert P and ET of April month into kg/m$^2$ Eq. 5 is used.

$$P \ or \ ET \ (m) = \left[ P \ or \ ET \left( \frac{kg}{m^2 s} \right) \times 10800 \left( \frac{s}{3hr} \right) \times 8 \left( \frac{3hr}{day} \right) \times 30 \ (days) \right] \times 0.001 (mm \ to \ m) \tag{5}$$

R (sum of Qs_acc, Qsb_acc and Qsm_acc) is measured in kg/m$^2$. Where Qs_acc, Qsb_acc and Qsm_acc
represents storm surface runoff, baseflow-groundwater runoff and snow melt. For accumulated
variables, monthly mean runoff is the average 3-hour accumulation. Thus, Eq. 6 is used to convert unit
of R.

$$R \ (m) = \left[ R \left( \frac{kg}{m^2 3hr} \right) \times 8 \left( \frac{3hr}{day} \right) \times 30 \ (days) \right] \times 0.001 (mm \ to \ m) \tag{6}$$

Please note that square bracket portion in Eq. 5 and 6 represents quantity in kg/m$^2$. And 1 kg/m$^2$ is equal
to 1 mm of P/ET/R. Thus, to convert mm to m factor of 0.001 is multiplied.

There exist a temporal lead or lag amongst the water budget components (P, ET and R). Hence, K (=12)
shifted time series of these datasets are generated and cyclostationary mean is removed. Consider a
mascon where total 36, 0.5°x0.5° grids are present. Our study period (January 2004 –December 2015)
contains total 144 months. Thus, data matrix dimensions for a given mascon are 144 (columns) x 36
(rows). To generate 12 months shifted time series of P, ET and R data from January 2003 is required.
When k=1, rows run from Jan 2003 to Dec 2014. For k=2, Jan 2003 gets replaced with Dec 2015, Feb
2003 changes to Jan 2003   and Dec 2014 changes to Nov 2014. In the end for k=12 rows run from Feb
2015 to Jan 2014. To remove cyclostationary mean, month wise average is computed for the period
between 2004 and 2015 resulting into 12 values which are removed from individual datasets. Thus, for
a single parameter after combining all 12 shifted timeseries matrix dimensions becomes 144 x 432.
Next, GRACE TWSA are detrended and annual cycle is removed to obtain residuals (144x36).
Combining P, ET, R and GRACE TWSA generates our final observation matrix with dimensions (144 x
(1332=432+432+432+36)). Which represents L in our regression model (Eq.1). Thereafter, residuals of
WGHM TWSA are obtained after removal of GRACE cyclo-stationary mean. This represents S in
equation 4.





Please note that entire operation is done at mascon scale unlike Vishwakarma et al. 2021 where they used catchment-scale. Thereafter, GRACE downscaled TWSA are obtained. And finally, trend and annual cycle are added back to the downscaled estimates.

## 3.3 Evaluation of downscaled products

First, validation of all three downscaled product is carried out using ref-GWSC measurements by computing correlation coefficient (r) and root-mean-square-error (RMSE). Secondly mascon wise performance of each approach is evaluated based on the improvement brought by new product w.r.t. original GRACE product. Thus, temporal gain metric (G) for r and RMSE is computed (Pascal et. al., 2022). And the formula is given below:

$$G = \frac{|M_{opt} - M_{LR}| - |M_{opt} - M_{HR}|}{|M_{opt} - M_{LR}| + |M_{opt} - M_{HR}|} \tag{7}$$

With $M_{LR}$ denotes the value of metric for Low resolution GRACE TWSA, $M_{HR}$ denotes the values of metric for downscaled product, and $M_{opt}$ the optimal value of this metric i.e., 1 for r and 0 for RMSE. Thus, gain in r ($G_r$) and RMSE ($G_{RMSE}$) are as follows:

$$G_r = \frac{|1 - r_{LR}| - |1 - r_{HR}|}{|1 - r_{LR}| + |1 - r_{HR}|} \tag{8}$$

$$G_{RMSE} = \frac{RMSE_{LR} - RMSE_{HR}}{RMSE_{LR} + RMSE_{HR}} \tag{9}$$

Whereas to quantify the effectiveness of approach in capturing the variability of WGHM model, gain in r w.r.t. WGHM is computed (Eq. 8). Mascons that show a positive gain in r and RMSE with respect to GRACE, as well as a positive gain in r with respect to WGHM, are considered reliable downscaling estimates. After comparing the products from three approaches, the approach which retained the maximum number of mascons after meeting these conditions is considered as the best approach.



**Figure 3.** Flow chart of methodology for downscaling GRACE TWSA and its evaluation in comparison to other existing downscaling products.



## 4 Results and discussions

### 4.1 Validation statistics for GRACE mascons

#### 4.1.1 Impact of surface water storage change contribution

Figure 4 illustrates the impact of surface water storage contribution (SWSC) on the validation statistics of GRACE groundwater storage change (GWSC) for mascons 27 (Govind Ballabh Pant reservoir contributes here) and 33 (Indira Sagar reservoir contributes here). The results show that GRACE GWSC (including SWSC) exhibited an excellent correlation with the reference GWSC (ref-GWSC), with r of 0.94 and 0.90 for mascons 27 and 33, respectively. Interestingly, removing SWSC did not lead to any significant improvement in r, with the values being 0.94 and 0.89, respectively.

In terms of RMSE, the values for both mascons were the same (0.13 m) when SWSC was not removed. After the removal of SWSC, a marginal improvement in RMSE was observed, with values of 0.12 m and 0.11 m for mascons 27 and 33, respectively. It is important to note that the overestimation of GRACE GWSC, when compared to ref-GWSC, is expected. This overestimation is likely due to the influence of other contributors to TWSA in GRACE, beyond just SWSC and soil moisture storage changes (SMSC).





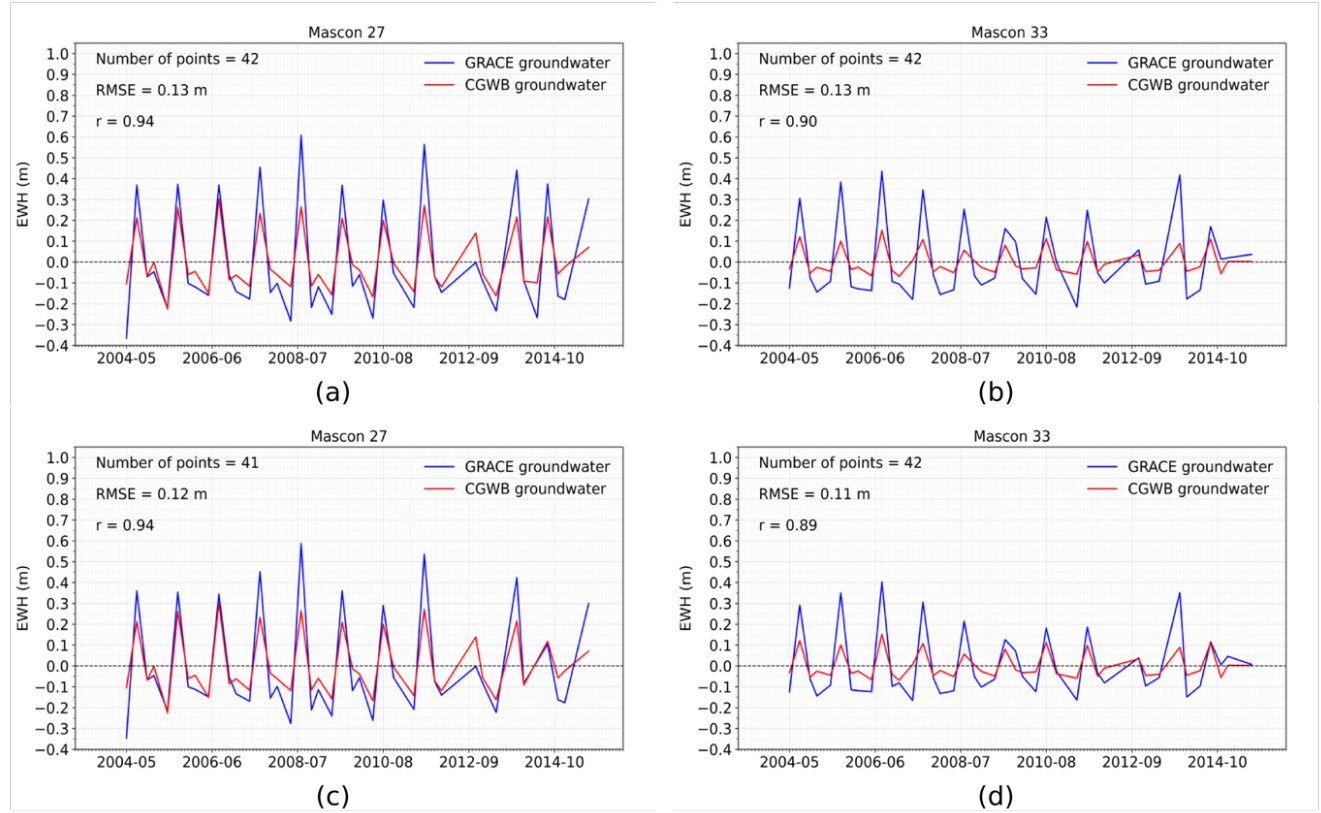

**Figure 4.** Comparison between GRACE derived groundwater storage changes and in-situ groundwater storage changes for a) Mascon 27 without removing surface water contribution. b) Mascon 33 without removing surface water contribution c) Mascon 27 after removing surface water contribution d) Mascon 33 after removing surface water contribution.

### 4.1.2 Validation statistics for GRACE mascons across India

Figure 5 indicates r and RMSE map of GRACE mascons GWSC over India when compared with ref-GWSC. The moderately good values of average r (0.75) and RMSE (0.13 m) are observed. Thus, GRACE mascons on an average are capable of explaining ~56 % variability in GWSC over India. These results are encouraging and confirm the potential of GRACE in capturing groundwater changes. The minimum r (0.13) is observed for mascon 5 covering Himachal Pradesh and maximum r (0.97) is observed for mascon 36 covering Odisha. Whereas the best RMSE (0.04 m) is observed for mascon





22(43) covering region from Gujarat (Andhra Pradesh and Odisha) and worst RMSE (0.28) is observed for mascon 16 covering majorly Uttar Pradesh and part of Madhya Pradesh.

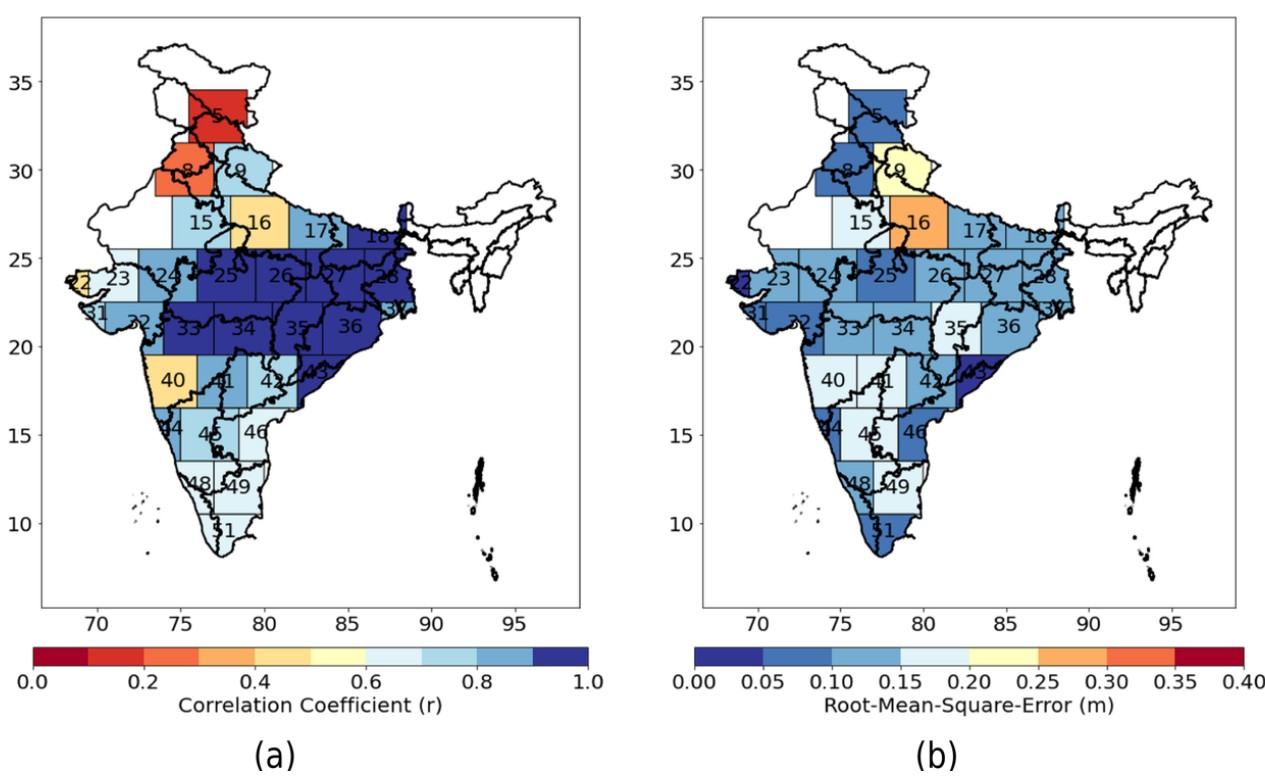

**Figure 5.** Mascon wise correlation and RMSE map of GRACE GWSC over India.

## 4.2 Validation and evaluation of downscaled products

Figure 6 indicates newly developed MMC downscaled product of GRACE JPL mascon TWSA of January 2004 over India. We observed that MMC downscaled product fails to work in a mascon 9 as method fails to generate coefficients and hence the downscaled product. Thus, to study whether shifting from catchment wise to mascon wise mass conservation, brings any significant improvement in the downscaled products, mascon 24 is selected and detailed analysis is given in section 4.2.1. The comprehensive statistics obtained for downscaled product over India are discussed in Section 4.2.2.





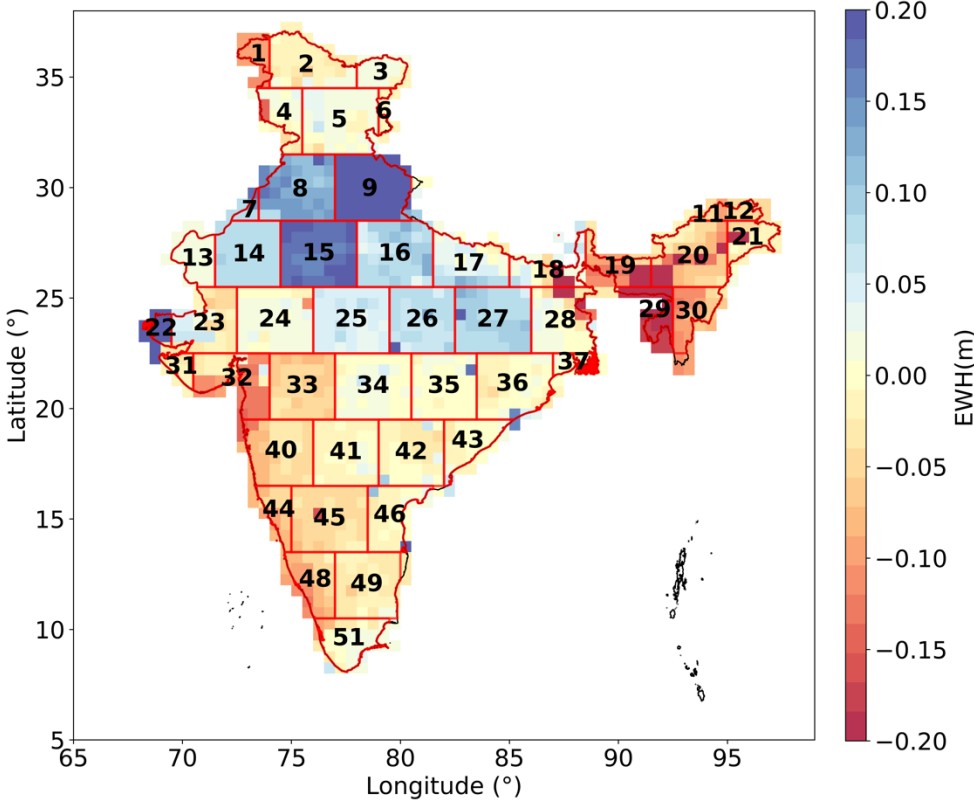

**Figure 6.** Downscaled (0.5°x0.5°) TWSA of January 2004 obtained from JPL mascon (3°x3°) using MMC approach over India. Numbers mentioned in the mascons refers to respective mascon number to facilitate mascon wise discussion.

**4.2.1 Comparison between MMC and CMC downscaled products: Mascon 24**

Traditionally, r and RMSE metrics are used to validate the parameter. For mascon 24, MMC approach showed evident improvement in r (Fig. 7b). Whereas CMC approach showed negative r (Fig. 7a). Figure 7c and d displays the temporal gain in r for each grid. This indicates MMC developed product shows downscaled grids with positive values (Fig. 7d) are more in comparison to CMC approach. Also, MMC approach shows small negative values (Fig. 7d). Whereas large negative gain values are indicated by CMC downscaled product (Fig. 7c).



**Figure 7.** GRACE mascon 24 grid wise correlation coefficient (r) of downscale groundwater storage product obtained using a) MMC method and b) Vishwakarma et al. (2021) when compared with in situ well measurement derived ground water storage changes.



The observed improvement can be attributed to mascon wise mass conservation approach. However, catchment wise approach does perform equally well for some other mascons. The performance of different approaches can be attributed to water holding capacity of region and consequently the aquifer type. There are studies which indicate that downscaling method performance is season dependent due to their dynamic and aquifer type (Pascal et al. 2022). However, for aquifers which are less sensitive to seasons, both approaches may work well.

### 4.2.2 Statistical comparison between MMC and CMC downscaled products over India

Figure 8 shows statistical comparison between both CMC and MMC downscaled products over India when validated w.r.t. ref-GWSC. The plot shows median values of r and RMSE obtained over mascons. Twenty-two mascons out of 52 are compared where GWL observations and 10 downscaled values are available. In case of MMC approach out of 22 mascons, 21 showed positive r whereas CMC resulted into 16 mascons with positive r. Indeed, MMC approach showed improved results over mascon 28, 32, 36, 46 and 51 by converting their r value from negative to positive (Fig. 8 a and c). Mascon 8 showed negative r in both the approaches. Overall reduction in variability of r values is also observed in MMC approach. Both products showed similar improvement in RMSE values (Fig. 8 b and d).





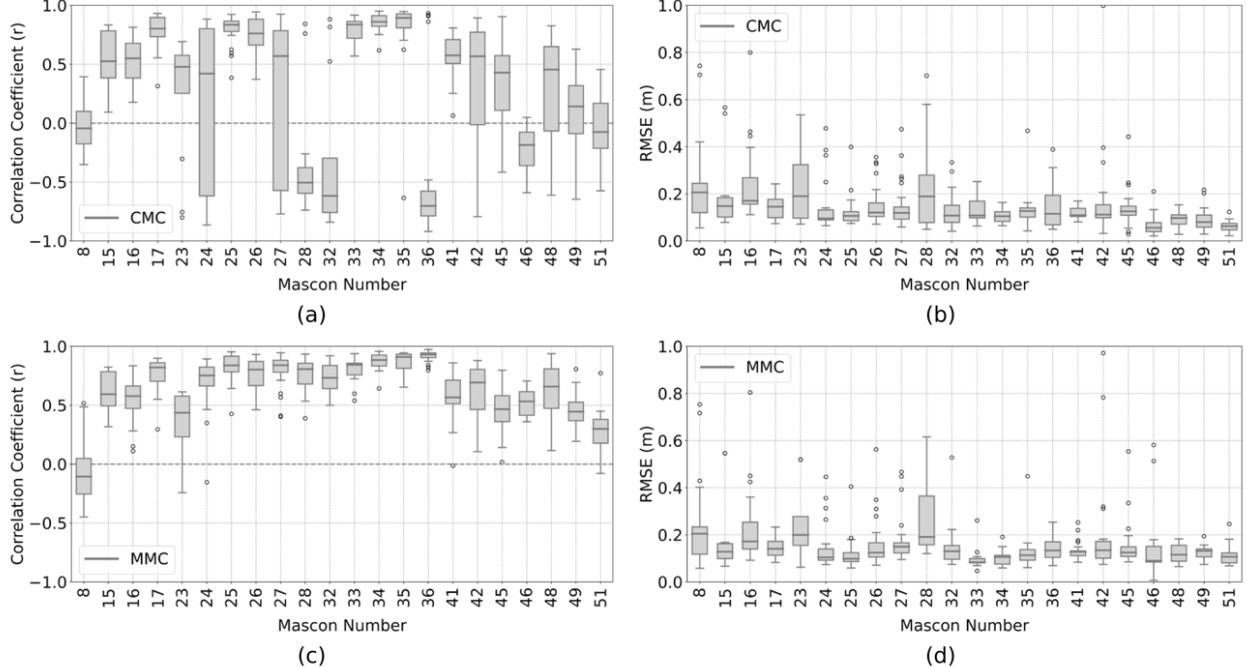

**Figure 8.** Box plots for (a) Correlation Coefficient of CMC product (b) RMSE of CMC product (c) Correlation Coefficient of MMC product (d) RMSE of MMC product over India.

340

Figure 9 demonstrates gain in r and RMSE w.r.t. GRACE computed for CMC and MMC approaches. The MMC approach showed positive gain in r for a greater number of mascons in comparison to CMC approach (Fig 9a, c). Indeed, CMC approach showed positive gain in r for only 3 mascons (17, 35 and 42) with maximum reaching to 7.98% for mascon17 whereas for MMC approach mascons with positive

345    gain are 12 in number (17, 24, 25, 26, 27, 33, 34, 35, 46, 48, 49 and 52) with maximum reaching to 8.85% for mascon 35. Also, MMC product shows lesser variability in values of gain in r.

Gain in RMSE shows similar performance in terms of median values. But reduced variability in values of gain in RMSE for MMC approach is also evident in Fig. 9d. Overall it is observed that MMC approach is outperforming the CMC approach as far as improvement w.r.t. original GRACE resolution

350    is concerned.





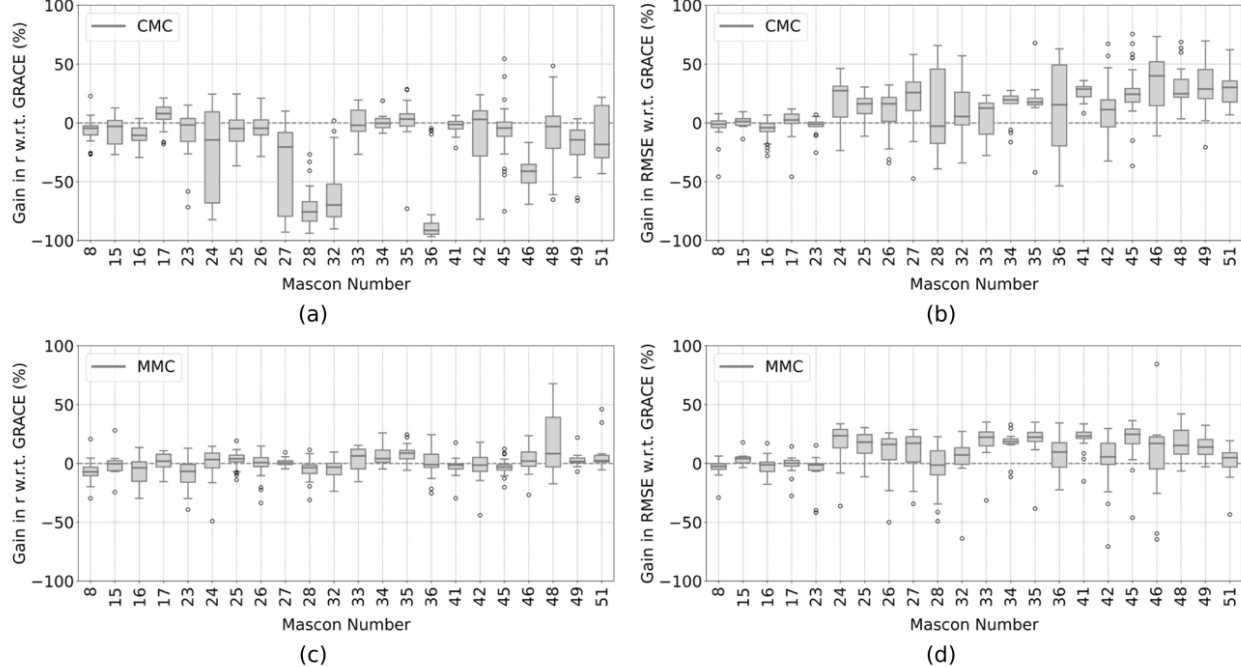

**Figure 9.** Box plots for gain in (a) Correlation Coefficient of CMC product (b) RMSE of CMC product (c) Correlation Coefficient of MMC product (d) RMSE of MMC product w.r.t. GRACE over India.

Figure 10 shows gain in r w.r.t. WGHM for CMC and MMC approaches. CMC approach showed positive gains for 12 out of 22 mascons along with large variability in it. But MMC approach showed positive gains for all mascons except one (mascon 8). This result also suggests that MMC approach is performing better in terms of mimicking the model variability in comparison to CMC approach.



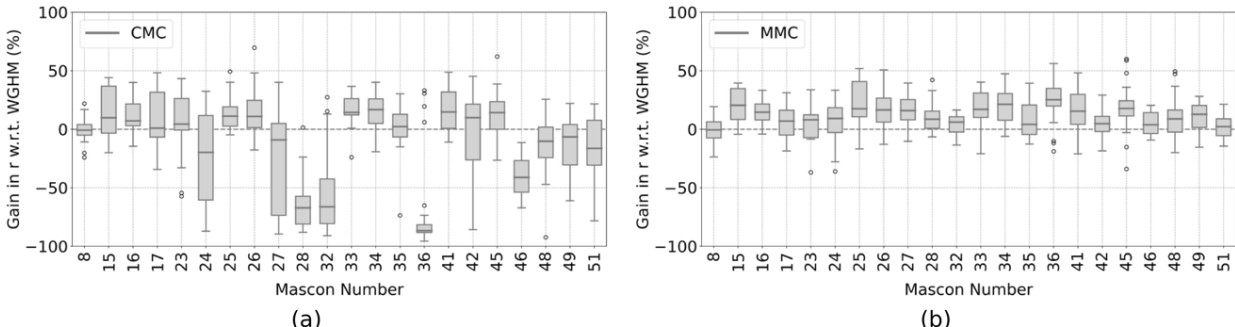

**Figure 10.** Box plots for gain in (a) Correlation Coefficient of CMC product (b) Correlation Coefficient of MMC product w.r.t WGHM over India.

Figure 11. displays r, RMSE, gain in r, gain in RMSE w.r.t. GRACE and gain in r w.r.t. WGHM for DL approach. It is observed that positive r values obtained for all 21 mascons with lower variability similar to that of MMC approach. Also, positive gains in r w.r.t. GRACE are observed for 10 mascons with maximum gain reaching to 9.68% for mascon 48. Gain in r w.r.t. WGHM is also observed to be positive for all mascons. Indicating DL also performs better in comparison to CMC approach.

Overall, it is observed that MMC approach is producing reliable downscaled product over more mascons in comparison to CMC and DL. Implementing reliable downscaled product criteria MMC approach retained 12 mascons whereas CCM and DL approach showed 3 and 10 masons, respectively suggesting MMC is outperforming CCM and DL. However, it is also observed that MMC is not able to produce downscaled product for some mascons despite that MMC found to be a superior downscaled product that produces more reliable estimates at fine resolution over a greater number of mascons.






**Figure 11.** Box plots of (a) Correlation Coefficient (b) RMSE (c) Gain in r w.r.t GRACE (d) Gain in RMSE w.r.t GRACE (e) Gain in r w.r.t. WGHM for DL product over India.

## 5 Conclusions

In this study we carried out validation and downscaling of GRACE observations over India. For
validation, contributions from major compartments of water column are considered i.e., surface water
changes, soil water changes and ground water changes. We conclude that GRACE at its native




resolution (about 3°x3°) when compared with in-situ groundwater well data, successfully captures the observed variations with excellent r and RMSE values. Implementing modified downscaling approach, a new downscaled product (MMC) is developed and compared with CMC. Over some mascon MMC
approach showed drastic improvement over CMC approach. Indeed, the CMC approach over some grids indicated strong negative r whereas the MMC approach showed moderate positive r. Thus, we found that MMC approach performs better in comparison to CMC approach. Next, we validated the other existing downscaled product (DL) and found that it also outperforms the CMC approach. Overall, it is concluded that MMC is outperforming the existing downscaled (CMC and DL) product and capable
of providing more reliable information of water storage changes across India at fine resolution.

**Data and Code Availability**

The downscaled TWSA product generated in this study is publicly available at: https://figshare.com/articles/dataset/GRACE_downscaled_TWSA_product_using_Mascon_wise_Mass_ Conservation/29196617. The raw data used in this study are publicly available. The JPL GRACE mason
product is available at https://grace.jpl.nasa.gov/. The WGHM TWSA simulations are available at https://doi.org/10.1594/PANGAEA.948461. The CDS soil moisture product is available at https://cds.climate.copernicus.eu. The CGWB data is available at https://indiawris.gov.in/wris/#/.

**Author contributions**

BDV conceived and supervised the study. MRS processed the data and performed the experiments and
drafted the manuscript. KSK contributed in developing example dataset. AS developed validation algorithm at initial steps. All authors contributed to the analysis of the results and commented on the manuscript.

**Competing interests**

The contact author has declared that none of the authors has any competing interests.



## Acknowledgements

We thank ISRO STC program for their generous financial support, the STC cell at IISc and SAC team for continuous support. We are extremely thankful to India-WRIS CGWB for giving access to data.

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
