# Peer review of "On the efficacy of downscaled GRACE total water storage products"

_EGUsphere, 2025_

## Author Comment (AC1)

**Response to reviewer's comments**

**# Reviewer 2**

We thank the reviewer for their comments. Our responses to all the comments are provided below.

**Initial summary comment:**

**Strengths:**

The manuscript by Suryawanshi et al. represents a strong effort to develop and validate a downscaling method for GRACE total water storage (TWS) anomalies over India. The authors propose a modified approach, called "Mascon-wise Mass Conserved" (MMC), building upon the methodology of Vishwakarma et al. (2021). The primary innovation lies in applying a mass conservation constraint at the native GRACE mascon scale (~3°x3°), rather than at the coarser catchment scale used in previous studies. A notable strength of the study is the large-scale validation framework, which integrates both statistical modeling and data assimilation within a hydrological context. The authors also compare MMC with previously published downscaling approaches, namely CMC (Vishwakarma et al., 2021) and DL (Gou and Soja, 2024). For this evaluation, they used the temporal "gain" metric introduced in a previous GRACE downscaling study in India (Pascal et al., 2022), which represents a positive step toward standardizing validation practices. The public release of the resulting dataset is also a valuable contribution to the community.

**Weaknesses:**

Despite these strengths, my principal concern relies on the hydrological validity of the downscaled product released. The manuscript lacks an appropriate discussion of the different hydrogeological contexts throughout India: huge irrigation, hydrogeological diversity, large dams and rivers, and small surface reservoirs simultaneously filling up with the monsoon and emptying with the dry season.

**Response:** We sincerely thank the reviewer for recognizing the strengths of our study, including the development of the Mascon-wise Mass Conserved (MMC) approach, the large-scale validation framework, comparison with previous methods (CMC and DL), use of the temporal *gain* metric, and the public release of the dataset. We greatly appreciate the positive assessment of these aspects.

We also thank the reviewer for highlighting the need to address the hydrological validity of the downscaled product across diverse contexts in India. In the revised manuscript, we will include a detailed discussion of different hydrogeological settings, including intensive irrigation areas, varied geological conditions, the influence of large dams and rivers, and small surface reservoirs that fill during the monsoon and drain during the dry season.

**Comment 1: Missing downscaling methods and validation state of the art:** The introduction does not adequately position the study relative to the state of the art, notably by failing to cite Pascal et al. (2022) as a framework for spatial and temporal validation of downscaling approaches in this specific Indian highly irrigated context. Missing downscaling studies in India should be discussed (Jyolsna et al., 2021 and Karunakalage et al., 2021). Similarly, the manuscript does not incorporate the full spatial and temporal validation approach proposed in Pascal et al., 2022. Should be justified or discussed somewhere.

**Response 1:** We appreciate the reviewer's comment. In the revised manuscript, we will strengthen the introduction by positioning our study relative to the current state of the art, including downscaling studies in India (e.g., Jyolsna et al., 2021; Karunakalage et al., 2021 and other).

I.    We will do the following modification in introduction and will insert table of reference discussed in the introduction.

TWSA from GRACE is difficult to validate due to unavailability of in-situ TWSA data (Scanlon et al., 2016) but not impossible if one compartment of TWSA is changing rapidly while others stay stable, especially over India where groundwater decline is alarming (Sarkar et al., 2020).  Another major challenging task to deal with is a coarse spatial resolution of GRACE data. Several hydrological, and agricultural studies demand regional/local scale inputs. To cater this demand several attempts to downscale GRACE TWSA have been conducted. Broadly those downscaling methods are categorised in two types: model-based/dynamic and data-based/statistical method. The model-based/dynamic approach is physically based, strongly depends on boundary conditions and computationally expensive (Schumacher et al., 2018; Sun et al., 2023). Whereas, in the data-based/statistical approach an empirical relationship is developed between coarse scale variables and fine scale variables. Owing to its computational efficiency and ease of implementation, the model-based/statistical approach has gained popularity among researchers. Related studies are summarized in Table. The statistical approach has been primarily implemented in three ways: simple linear regression, multivariate linear regression, and machine learning algorithms. In simple linear regression, a single variable is regressed against GRACE TWSA. For instance, Gemitzi et al. (2021) and Yin et al. (2018) used only precipitation and evapotranspiration, respectively, to downscale GRACE data, as these were identified as the dominant drivers in their respective regions. To address cases where a single variable is insufficient to explain TWSA variability, multivariate linear regression with a water budget constraint has been applied (Karunakalage et al., 2021; Ning et al., 2014; Vishwakarma et al., 2021). To account for nonlinearity, researchers have also implemented machine learning algorithms such as random forest, artificial neural networks, and long short-term memory etc. (Ali et al., 2021; Arshad et al., 2025; Chen et al., 2019; Gorugantula and Kambhammettu, 2022; Gou and Soja, 2024; Jyolsna et al., 2021; Kalu et al., 2024; Miro and Famiglietti, 2018; Pascal et al., 2022). Despite ongoing efforts to enhance the spatial resolution of GRACE data, continued advancements in data processing techniques and the availability of improved data products offer substantial scope for further improvement.

**Table.** Summary of the statistical downscaling studies applied to GRACE data.

| Reference | Statistical downscaling method | Original resolution | Downscaled resolution | Study region |
|---|---|---|---|---|
| **Simple linear regression** | | | | |
| Gemitzi et al. (2021) | Simple linear regression using GPM-IMERG precipitation | 1° × 1° | 0.1° × 0.1° | Greece |
| Yin et al. (2018) | Simple linear regression using ET | 110 × 110 km | 2 × 2 km | North China Plain |
| **Multivariate linear regression** | | | | |
| Karunakalage et al. (2021) | Multivariate regression model with water budget closure constraint | 1° × 1° | 0.25° × 0.25° | Mehsana district, Gujarat, India |
| Ning et al. (2014) | Multivariate regression model with water budget closure constraint | 1° × 1° | 0.25° × 0.25° | Yunnan province, China |
| Vishwakarma et al. (2021) | Multivariate regression model with water budget closure constraint | 3° × 3° | 0.5° × 0.5° | Global |
| **Machine learning algorithms** | | | | |

| | | | | |
|---|---|---|---|---|
| Ali et al. (2021) | Artificial neural network and Random Forest | 1° × 1° | 0.25° × 0.25° | Indus basin irrigation system |
| Arshad et al. (2025) | Random forest, CART, Gradient tree boosting algorithms | 55 × 55 km | 1× 1 km | Saudi Arabia |
| Chen et al. (2019) | Random Forest | 1° × 1° | 0.25° × 0.25° | Northeast of mainland China |
| Gorugantula and Kambhammettu (2022) | Long Short-Term Memory | 1° × 1° | 0.25° × 0.25° | Krishna River Basin, India |
| Gou and Soja (2024) | Convolutional eural network | 3° × 3° | 0.5° × 0.5° | Global |
| Jyolsna et al. (2021) | Multi variate regression, Random Forest | 1° × 1° | 0.25° × 0.25° | Four contrasting hydrogeological basins of India |
| Kalu et al. (2024) | Support Vector Machine (SVM) with water budget closure constraint | 1° × 1° | 0.25° × 0.25° | Northern Australia (the Cambrian Limestone Aquifer—CLA) |
| Miro and Famiglietti (2018) | Artificial neural network | 2,00,000 km² | 16 km² | California's Central Valley |
| Pascal et al. (2022) | Multi linear regression, Random Forest | 3° × 3° | 0.5° × 0.5° | A fractured crystalline aquifer in southern India |

II.     Section 3.3 will be modified to explain why only the temporal gain is implemented and not the spatial gain from Pascal et al. (2022).

**3.3 Evaluation of downscaled products**

First, validation of all three downscaled products is performed using Ref-GWSC measurements by computing the Correlation Coefficient (r) and Root-Mean-Square Error (RMSE). In addition to these classical evaluation metrics, the temporal metric gain proposed by Pascal et al. (2022) is implemented to compare the performance of the downscaled products. As noted by Pascal et al. (2022) most studies do not assess the performance of downscaled GRACE products relative to the original GRACE data. By using the temporal gain metric, we can quantitatively evaluate the accuracy of the downscaled products.

In the present study, GRACE JPL mascon data are used as the low-resolution reference without applying the scale factor. These scale factors, being multiplicative and model-based, can amplify any uncertainties in the GRACE product if the model does not align with reality. Vishwakarma et al. (2017) and Pascal et al. (2022) have shown that applying the scale factor can degrade rather than improve the results. Consequently, we have not extended the validation framework to compute spatial gain, since our low-resolution reference does not

incorporate the scale factor and thus does not contain spatial variability. As illustrated in Figure, the reference values are uniform over each mascon.

[Figure]

**Figure x.** GRACE mascons over India for September 2004. Red boxes represent mascon boundaries.

**Comment 2: Missing Table 2 Specific Yield of aquifers:** Methodologically, the transformation of groundwater levels into groundwater storage (GWS) relies on specific yield, yet Table 2, which should detail the assigned specific yield values for different aquifer types, is missing, limiting transparency and reproducibility. The choice of eight clusters for k-means clustering is arbitrary and unsubstantiated, raising concerns about the spatial reliability of the reference dataset (Ref-GWSC). The assigned specific yield values appear disconnected from geologically coherent units, and the exclusive use of temporal gain for validation neglects the assessment of spatial variability within mascons, which is a fundamental objective of downscaling.

**Response 2:** We thank the reviewer for these important comments. In the revised manuscript, we will address these points as follows:

1. We will include hydrogeological map (Figure and Table ) (Bhanja et al., 2016) as shown below to clearly list the assigned specific yield values for different aquifer types, ensuring transparency and reproducibility of the groundwater storage calculation.

[Figure]

**Figure.** Hydrogeology map of India (Kuruva et al., 2025)

**Table.** Specific yield values for varying hydrogeologic setting shown Figure x (Source: Bhanja et al., 2016)

| S.No | Hydrogeology | $S_y$ range | Mean $S_y$ |
|------|--------------|-------------|------------|
| 1 | Unconsolidated sedimentary | 0.06 to 0.20 | 0.130 |
| 2 | Consolidated, permeable sedimentary | 0 to 0.08 | 0.043 |
| 3 | Sedimentary aquitards | 0 to 0.03 | 0.018 |
| 4 | Folded metasediments/metamorphics | 0 to 0.03 | 0.018 |
| 5 | Jointed crystalline | 0.01 to 0.03 | 0.020 |
| 6 | Fractured crystalline | 0 to 0.04 | 0.023 |

2.  We acknowledge that the choice of eight clusters for k-means clustering may appear arbitrary. We recognized that the use of k-means clustering on RGB values introduces unnecessary uncertainty and risks misclassifying hydrogeological units due to which we will replace this approach with a vectorization-based method using Quantum Geographic Information System (QGIS) as adopted by Kuruva et al. (2025). We will modify methodology as follows:

**3.1.1 Conversion of GWL changes to groundwater storage changes**

To compute GWSC (m EWH), quality controlled GWL change (GWLC) is multiplied with Specific Yield (SY). Where SY is a dimensionless factor that indicates the fraction of total ground water volume that would yield under gravity and is used to convert water level to water storage. Fundamentally it is a hydrogeological property of an aquifer.

These SY values are extracted from hydrogeology map of India (Figure x and Table x. Source: Bhanja et al., (2016)) at quality-controlled well locations using vectorization method in Quantum Geographic Information System (QGIS) platform. The required raster layer of hydrogeology map is downloaded from https://doi.org/10.6084/m9.figshare.29293877.v3 (Kuruva et al., 2025). Then quality controlled GWLs are converted into point shape file and overlaid on this raster layer to extract SY values at each well location using "sample raster values" tool in QGIS. The obtained SY is multiplied with quality controlled GWLC to get reference groundwater storage changes referred as Ref-GWSC.

3.  We didn't extend the validation framework by incorporating the spatial gain metric as we are using GRACE JPL mascon as it is (i.e., without multiplying with scale factor) as our low-resolution reference in the present study. These scale factors are computed using a model and are multiplicative in nature, which means any uncertainty in GRACE product will be inflated by the scale factor if the model does not agree with the truth. Vishwakarma et al., 2017 has shown that scale factor method can do more damage than good. The mascons over India for September 2004 is shown below. The Figure demonstrates that there is no spatial variability within individual mascons. Therefore, the spatial gain metric is not incorporated in this study.

[Figure]

**Figure.** GRACE mascons over India for September 2004. Red boxes represent mascon boundaries.

**Comment 3: Surface water fluctuation neglected:** The validation relies on well-known Central Ground Water Board groundwater levels across 22 mascons, showing that native GRACE explains only 56% of in situ GWS variability. This implies that the remaining variability arises from surface and soil water storage, a well-documented phenomenon in both the Ganges-Brahmaputra basin (Salameh et al., 2017) and southern India, particularly Telangana. In Telangana, the cumulative capacity of large reservoirs on main rivers is estimated at 113 mm of equivalent water height, and small upstream reservoirs contribute an additional 30 mm, representing approximately 24% of the TWS signal over the period 2002–2021 (Pascal et al., 2021). This potential reservoir capacity of 143 mm represents about 24% of the annual GRACE TWS fluctuation in the area during 2002–2021 (600 mm). However, the authors interpretation that reservoirs are "rarely simultaneously full" is misleading: under the monsoonal regime, filling generally occurs simultaneously during the wet season and drawdown during the dry season as well. The decision to neglect this contribution can only be justified on practical grounds, namely the difficulty of obtaining reliable regional data on the filling dynamics of both large and small reservoirs, but not on theoretical arguments about negligible impact. Suryawanshi et al. consider surface water negligible based on tests over only two reservoirs, and they appear to generalize these results to the entire country, which is an overextension.

**Response 3:** We thank the reviewer for this insightful comment. In the revised manuscript, we will explicitly acknowledge the role of available surface water bodies as well as other components of water storages using WGHM (WaterGAP Hydrology Model) output. This TWS product comprises of various water storages such as canopy water storage, snow water equivalent, soil moisture, river streams, lakes, reservoir, wetlands and groundwater. To align with GRACE TWSA base line, we generated TWSA from WGHM by removing mean of baseline period (2004 to 2009). In addition, separate ground water storage product from WGHM is also downloaded and subtracted from WGHM TWS to isolate the water storages due to canopy, snow, soil moisture, river streams, lakes, reservoirs, and wetlands. This isolated water storage component is further used to sperate the groundwater storage component from TWSA products used in the study. Based on this validation methodology will be revised. For instance section 4.1 will be modified as follows and subsequently other sections will also be modified.

**4.1 Validation statistics for GRACE GWSC at native resolution (3° × 3°) across India**
Figure indicates mascon-wise maps of r and RMSE between GRACE GWSC and ref-GWSC across India. The validation is performed only for 36 mascons out of 52 owing to missing well observations. Out of 36, 28 mascons showed positive r ranging from 0.06 to 0.91. Mascons 5 and 27 showed a maximum r value of 0.91, indicating

that GRACE is capable of explaining 83% of GWSC variability in these mascons. Mascons 19, 26, 35, 36 and 41 showed r values varying between 0.83 to 0.86 explaining the variability of GWSC between 69% to 74%. For mascons 9, 10, 15, 18, 24, 25, 28, 34, 37, 40, 45, 49 r is ranging between 0.63 and 0.79, explaining 40 to 60% of GWSC variability. These results indicates that GRACE GWSC is capable of explaining variability in GWSC over 40% for majority of the mascons. In contrast, mascons 16, 32, 43, and 48 showed low r values (0.15, 0.06, 0.17 and 0.10, respectively), indicating that GRACE is not even capturing a 3% of GWSC variability. Mascons 4, 8, 17, 22, 23, 44 and 50 exhibited negative correlations. RMSE values observed to be ranging between 0.06 m to 0.32 m. The best (minimum) RMSE is observed for mascon 6 whereas worst (maximum) is shown by mascon 16. Overall RMSE remained low for majority of the mascons (< 0.16 m) whereas mascon 9, 16, 17 showed higher RMSE varying between 0.29 m and 0.32 m.

Interestingly, we also found that the validation performance is independent of number and distribution of wells in the mascon. For instance, mascon 17, with 112 wells, exhibited a negative correlation of -0.32, whereas mascon 5, with only 33 wells showed maximum correlation of 0.91. Similarly, mascon 15, where all wells are clutered in the right side of the mascon, still showed a good correlation of 0.63. Whereas mascon 16 has a fairly well distributed network of wells, still showed a poor correlation of 0.15. Thus, the failure of GRACE in capturing groundwater signal in some mascons can be attributed to multiple factors such as, errors in measurements of GWL, high irrigational activities occurring at local scales, inadequacy of WGHM model in computing exact storages of canopy, snow, soil moisture, river streams, lakes, reservoirs, and wetlands. Conversely, in some mascons GRACE performs exceptionally well, achieving r = 0.91 and RMSE = 0.09 m. The strong agreement also underscores the superiority of the difference-based validation method, as it preserves the integrity of observed data and yields more reliable results than approaches that rely on temporal interpolation of CGWB measurements.

[Figure]

**Figure.** Mascon-wise maps of (a) the correlation coefficient (r) and (b) RMSE between GRACE GWSC and ref-GWSC over India.

**Comment 4:** Deconvolution problem for validation with the ref-GWSC: Consequently, the downscaling approach risks misattributing surface water fluctuations, due to reservoir filling and releases, to groundwater storage. By considering only surface soil moisture (~5 cm) and neglecting the majority of reservoirs, the study likely

overestimates GWS, and the signal attributed to GWS becomes a hybrid of groundwater and surface water. Such misinterpretation can distort seasonal dynamics, for instance by incorrectly suggesting rapid aquifer recharge when TWS increases during monsoon reservoir filling. Overall, the study underestimates the importance of surface water in TWS deconvolution, and the resulting GWS product may therefore be systematically biased, which leads to an impossibility to validate this published downscaled TWS dataset.

**Response 4:** We thank the reviewer for this detailed and important comment. We acknowledge that by considering only surface soil moisture (~5 cm) and omitting most reservoirs, the downscaled product may partially conflate groundwater storage (GWS) with surface water fluctuations, potentially biasing seasonal dynamics. As mentioned in "Response 4" we will be using WGHM TWS products. In this product soil moisture storage is computed up to root zone depth ranging between 0.1 m to 4 m, which provides a more realistic representation of subsurface storage dynamics. Furthermore, the inclusion of surface water components such as rivers, lakes, wetlands, and reservoirs in WGHM will help reduce the risk of misattributing surface water variability to groundwater. This refinement should improve the physical consistency of the downscaled TWS–GWS separation and enhance the reliability of the validation against the ref-GWSC dataset.

*Please note that additionally, in the revised manuscript, we will extend the Mascon-wise Mass Conservation (MMC) downscaled product up to December 2023, thereby improving the temporal coverage of the dataset.*

**References cited in this response:**

Ali, S., Liu, D., Fu, Q., Cheema, M. J. M., Pham, Q. B., Rahaman, M. M., Dang, T. D., and Anh, D. T.: Improving the Resolution of GRACE Data for Spatio-Temporal Groundwater Storage Assessment, Remote Sens., 13, 3513, https://doi.org/10.3390/rs13173513, 2021.

Arshad, A., Shafeeque, M., Tran, T. N. D., Mirchi, A., Xiang, Z., He, C., AghaKouchak, A., Besnier, J., and Rahman, M. M.: Multi-model ensemble machine learning-based downscaling and projection of GRACE data reveals groundwater decline in Saudi Arabia throughout the 21st century, J. Hydrol.: Reg. Stud., 60, 102552, https://doi.org/10.1016/j.ejrh.2025.102552, 2025.

Bhanja, S. N., Mukherjee, A., Saha, D., Velicogna, I., and Famiglietti, J., S.: Validation of GRACE based groundwater storage anomaly using in-situ groundwater level measurements in India, J. Hydrol., 543, 729-738, https://doi.org/10.1016/j.jhydrol.2016.10.042, 2016.

Chen, L., He, Q., Liu, K., Li, J., and Jing, C.: Downscaling of GRACE-Derived Groundwater Storage Based on the Random Forest Model, Remote Sens., 11, 2979, https://doi.org/10.3390/rs11242979, 2019.

Gemitzi, A., Koutsias, N., and Lakshmi, V.: A spatial downscaling methodology for GRACE total water storage anomalies using GPM IMERG precipitation estimates, Remote Sens., 13, 5149, https://doi.org/10.3390/rs13245149, 2021.

Gorugantula, S. S., and Kambhammettu, B. P.: Sequential downscaling of GRACE products to map groundwater level changes in Krishna River basin, Hydrolog. Sci. J., 67, 1846-1859, https://doi.org/10.1080/02626667.2022.2106142, 2022.

Gou, J., and Soja, B.: Global high-resolution total water storage anomalies from self-supervised data assimilation using deep learning algorithms, Nature Water ,2, 139-150,https://doi.org/10.1038/s44221-024-00194-w, 2024.

Jyolsna, P. J., Kambhammettu, B. V. N. P., and Gorugantula, S.: Application of random forest and multi-linear regression methods in downscaling GRACE derived groundwater storage changes, Hydrolog. Sci. J., 66, 874–887, https://doi.org/10.1080/02626667.2021.1896719, 2021.

Kalu, I., Ndehedehe, C. E., Ferreira, V. G., Janardhanan, S., Currell, M. and Kennard, M. J.: Statistical downscaling of GRACE terrestrial water storage changes based on the Australian Water Outlook model. Sci. Rep., 14, 10113, https://www.nature.com/articles/s41598-024-60366-2, 2024.

Karunakalage, A., Sarkar, T., Kannaujiya, S., Chauhan, P., Pranjal, P., Taloor, A. K., and Kumar, S.: The appraisal of groundwater storage dwindling effect, by applying high resolution downscaling GRACE data in and around Mehsana district, Gujarat, India, Groundwater Sustain. Dev., 13, 100559, https://doi.org/10.1016/j.gsd.2021.100559, 2021.

Kuruva, S. K., Suryawanshi, M. R., Shakya, A., Va, C., Shaw, B., Sukumaran, V., Kour, R., Kochar, A., Chander, S., Nikam, B. R., and Dasika, N. K.: Quality controlled, reliable groundwater level data with corresponding specific yield over India, Scientific Data, 12, 1609, https://doi.org/10.1038/s41597-025-05899-5, 2025.

Miro, M. E., and Famiglietti, J. S.: Downscaling GRACE remote sensing datasets to high-resolution groundwater storage change maps of California's Central Valley. Remote Sens., 10, 1-18, https://doi.org/10.3390/rs10010143, 2018.

Ning, S., Ishidaira, H., and Wang, J.: Statistical Downscaling of Grace-Derived Terrestrial Water Storage Using Satellite and Gldas Products, J. Jpn. Soc. Civ. Eng. Ser. B1, 70, I_133–I_138, https://doi.org/10.2208/jscejhe.70.I_133, 2014.

Pascal, C., Ferrant, S., Selles, A., Maréchal, J.-C., Paswan, A., and Merlin, O.: Evaluating downscaling methods of GRACE (Gravity Recovery and Climate Experiment) data: a case study over a fractured crystalline aquifer in southern India, Hydrol. Earth Syst. Sci., 26, 4169–4186, https://doi.org/10.5194/hess-26-4169-2022, 2022.

Sarkar, T., Kannaujiya, S., Taloor, A. K., Ray, P. K. C. and Chauhan, P.: Integrated study of GRACE data derived interannual groundwater storage variability over water stressed Indian regions. Groundwater Sustain. Dev., 10, 100376, https://doi.org/10.1016/j.gsd.2020.100376, 2020.

Scanlon, B. R., Zhang, Z., Save, H., Wiese, D. N., Landerer, F. W., Long, D., Longuevergne, L., and Chen, J.: Global evaluation of new GRACE mascon products for hydrologic applications, Water Resour. Res., 52, 9412-9429, https://doi.org/10.1002/2016WR019494, 2016.

Schumacher, M., Forootan, E., van Dijk, A. I. J. M., Müller Schmied, H., Crosbie, R. S., Kusche, J., and Döll, P.: Improving drought simulations within the Murray-Darling Basin by combined calibration/assimilation of GRACE data into the WaterGAP Global Hydrology Model, Remote Sens. Environ., 204, 212–228, https://doi.org/10.1016/j.rse.2017.10.029, 2018.

Sun, J., Hu, L., Cao, X., Liu, D., Liu, X. and Sun, K.: A dynamical downscaling method of groundwater storage changes using GRACE data, J. Hydrol.: Reg. Stud, 50, 101558, https://doi.org/10.1016/j.ejrh.2023.101558, 2023.

Vishwakarma, B. D., Horwath, M., Devaraju, B., Groh, A., and Sneeuw, N.: A data-driven approach for repairing the hydrological catchment signal damage due to filtering of GRACE products, Wat. Resour. Res., 53, 9824-9844, https://doi.org/10.1002/2017WR021150, 2017.

Vishwakarma, B. D., Zhang, J., and Sneeuw, N.: Downscaling GRACE total water storage change using partial least squares regression, Scient. Data, 8, 95, https://doi.org/10.1038/s41597- 021-00862-6, 2021.

Yin, W., Hu, L., Zhang, M., Wang, J., and Han, S.-C.: Statistical Downscaling of GRACE-Derived Groundwater Storage Using ET Data in the North China Plain, J. Geophys. Res.-Atmos., 123, 5973–5987, https://doi.org/10.1029/2017JD027468, 2018.

---

## Author Comment (AC2)

**Response to reviewer's comments**

**# Reviewer 1**

We thank the reviewer for their comments. Our responses to all the comments are provided below.

**Initial summary comment:** The manuscript mainly examines the downscaling of GRACE mascon-based total water storage anomalies over India, introducing a mascon-scale mass conservation approach and validating the product with in-situ groundwater well observations. The topic is of broad interest to researchers in hydrology and remote sensing, and the dataset produced could be potentially useful for regional water resource studies. However, the study builds closely on previous work (Vishwakarma et al., 2021), and the novelty is therefore somewhat limited. In addition, the manuscript suffers from some shortcomings in the literature review, methodological justification, interpretation of the results, and presentation of figures and references. Therefore, the manuscript requires substantial revisions before it can be considered for publication.

**Response:** Thank you for the constructive feedback and acknowledging the relevance of our study in the field of hydrology and remote sensing, as well as the potential value of the produced downscaled product. Although the study is closely built on the work of Vishwkarma et al. (2021); the proposed Mascon-wise Mass Conservation (MMC) approach substantially improves the efficacy of downscaled product. Moreover, the validation performed in this study provides a robust assessment of the efficacy of downscaled products which was lacking in earlier studies. For this evaluation, we adopted the temporal gain metric proposed by Pascal et al. (2022), which is an important step towards standardizing validation practices. These factors (MMC and validation) have also been recognised by reviewer two as main strength and novelty of this manuscript.

We will substantially revise the manuscript to address all the comments. Some of the major changes will be:

- Expanding the literature survey (including the articles Jyolsna et al. 2021, Karunakalage et al. 2021: list of the articles will be added as table in the revised manuscript).
- Providing a more detailed and clearer justification of methodology.
- Improving the interpretation and discussion of results within different hydrogeological contexts across India.
- Improving the readability and presentation of figures and references.

**Major Comments**

**Comment 1:** [Introduction] The literature review does not sufficiently describe international developments in GRACE downscaling methods. The limited set of references makes it difficult to see how the study advances beyond existing work.

**Response 1:** We thank the reviewer for this valuable comment. We agree that the current version does not sufficiently cover international developments in GRACE downscaling research. In the revised manuscript, the literature review will be substantially expanded to include national and international studies (listed in the Table below). Following modified paragraph will be inserted in the manuscript:

TWSA from GRACE is difficult to validate due to unavailability of in-situ TWSA data (Scanlon et al., 2016) but not impossible if one compartment of TWSA is changing rapidly while others stay stable, especially over India where groundwater decline is alarming (Sarkar et al., 2020). Another major challenging task to deal with is a coarse spatial resolution of GRACE data. Several hydrological, and agricultural studies demand regional/local scale inputs. To cater this demand several attempts to downscale GRACE TWSA have been conducted. Broadly those downscaling methods are categorised in two types: model-based/dynamic and data-based/statistical method. The model-based/dynamic approach is physically based, strongly depends on boundary conditions and computationally expensive (Schumacher et al., 2018; Sun et al., 2023). Whereas, in the data-based/statistical approach an empirical relationship is developed between coarse scale variables and fine scale variables. Owing

to its computational efficiency and ease of implementation, the model-based/statistical approach has gained popularity among researchers. Related studies are summarized in Table. The statistical approach has been primarily implemented in three ways: simple linear regression, multivariate linear regression, and machine learning algorithms. In simple linear regression, a single variable is regressed against GRACE TWSA. For instance, Gemitzi et al. (2021) and Yin et al. (2018) used only precipitation and evapotranspiration, respectively, to downscale GRACE data, as these were identified as the dominant drivers in their respective regions. To address cases where a single variable is insufficient to explain TWSA variability, multivariate linear regression with a water budget constraint has been applied (Karunakalage et al., 2021; Ning et al., 2014; Vishwakarma et al., 2021). To account for nonlinearity, researchers have also implemented machine learning algorithms such as random forest, artificial neural networks, and long short-term memory etc. (Ali et al., 2021; Arshad et al., 2025; Chen et al., 2019; Gorugantula and Kambhammettu, 2022; Gou and Soja, 2024; Jyolsna et al., 2021; Kalu et al., 2024; Miro and Famiglietti, 2018; Pascal et al., 2022). Despite ongoing efforts to enhance the spatial resolution of GRACE data, continued advancements in data processing techniques and the availability of improved data products offer substantial scope for further improvement.

**Table.** Summary of the statistical downscaling studies applied to GRACE data.

| Reference | Statistical downscaling method | Original resolution | Downscaled resolution | Study region |
|---|---|---|---|---|
| **Simple linear regression** | | | | |
| Gemitzi et al. (2021) | Simple linear regression using GPM-IMERG precipitation | 1° × 1° | 0.1° × 0.1° | Greece |
| Yin et al. (2018) | Simple linear regression using ET | 110 × 110 km | 2 × 2 km | North China Plain |
| **Multivariate linear regression** | | | | |
| Karunakalage et al. (2021) | Multivariate regression model with water budget closure constraint | 1° × 1° | 0.25° × 0.25° | Mehsana district, Gujarat, India |
| Ning et al. (2014) | Multivariate regression model with water budget closure constraint | 1° × 1° | 0.25° × 0.25° | Yunnan province, China |
| Vishwakarma et al. (2021) | Multivariate regression model with water budget closure constraint | 3° × 3° | 0.5° × 0.5° | Global |
| **Machine learning algorithms** | | | | |
| Ali et al. (2021) | Artificial neural network and Random Forest | 1° × 1° | 0.25° × 0.25° | Indus basin irrigation system |
| Arshad et al. (2025) | Random forest, CART, Gradient tree boosting algorithms | 55 × 55 km | 1× 1 km | Saudi Arabia |
| Chen et al. (2019) | Random Forest | 1° × 1° | 0.25° × 0.25° | Northeast of mainland China |
| Gorugantula and Kambhammettu (2022) | Long Short-Term Memory | 1° × 1° | 0.25° × 0.25° | Krishna River Basin, India |

| | | | | |
|---|---|---|---|---|
| Gou and Soja (2024) | Convolutional eural network | 3° × 3° | 0.5° × 0.5° | Global |
| Jyolsna et al. (2021) | Multi variate regression, Random Forest | 1° × 1° | 0.25° × 0.25° | Four contrasting hydrogeological basins of India |
| Kalu et al. (2024) | Support Vector Machine (SVM) with water budget closure constraint | 1° × 1° | 0.25° × 0.25° | Northern Australia (the Cambrian Limestone Aquifer—CLA) |
| Miro and Famiglietti (2018) | Artificial neural network | 2,00,000 $km^2$ | 16 $km^2$ | California's Central Valley |
| Pascal et al. (2022) | Multi linear regression, Random Forest | 3° × 3° | 0.5° × 0.5° | A fractured crystalline aquifer in southern India |

**Comment 2:** [Methodology, Line 165] The use of k = 8 in the k-means clustering requires further justification. Please explain the reason for this choice and whether sensitivity tests were performed.

**Response 2:** Thanks for this insightful comment. We now recognize that the use of k-means clustering on RGB values introduces unnecessary uncertainty and risks misclassifying hydrogeological units. Hence, we will replace this approach with a vectorization-based method using Quantum Geographic Information System (QGIS) as adopted by Kuruva et al. (2025).

For this vectorization-based approach, the hydrogeology map is used in TIF format. Each polygon within this file is assigned with corresponding Specific Yield (SY) value as mention in Table 2 (Bhanja et al., 2016). This TIF file is then used to extract SY values for the quality-controlled groundwater well observations. Thus, following section will be inserted in methodology section of manuscript:

**3.1.1 Conversion of GWL changes to groundwater storage changes**

To compute GWSC (m EWH), quality controlled GWL change (GWLC) is multiplied with Specific Yield (SY). Where SY is a dimensionless factor that indicates the fraction of total ground water volume that would yield under gravity and is used to convert change in water level to change in water storage. Fundamentally it is a hydrogeological property of an aquifer.

These SY values are extracted from hydrogeology map of India (Figure and Table. Source: Bhanja et al., (2016)) at quality-controlled well locations using vectorization method in Quantum Geographic Information System (QGIS) platform. The required raster layer of hydrogeology map is downloaded from https://doi.org/10.6084/m9.figshare.29293877.v3 (Kuruva et al., 2025). Then quality controlled GWLs are converted into point shape file and overlaid on this raster layer to extract SY values at each well location using "sample raster values" tool in QGIS. The obtained SY is multiplied with quality controlled GWLC to get reference groundwater storage changes hereafter referred as Ref-GWSC.

[Figure]

**Figure.** Hydrogeology map of India (Kuruva et al., 2025)

**Table.** Specific yield values for varying hydrogeologic setting shown in Figure x (Source: Bhanja et al., 2016)

| S.No | Hydrogeology | $S_y$ range | Mean $S_y$ |
|------|--------------|-------------|------------|
| 1 | Unconsolidated sedimentary | 0.06 to 0.20 | 0.130 |
| 2 | Consolidated, permeable sedimentary | 0 to 0.08 | 0.043 |
| 3 | Sedimentary aquitards | 0 to 0.03 | 0.018 |
| 4 | Folded metasediments/metamorphics | 0 to 0.03 | 0.018 |
| 5 | Jointed crystalline | 0.01 to 0.03 | 0.020 |
| 6 | Fractured crystalline | 0 to 0.04 | 0.023 |

**Comment 3:** [Methodology] Some equations are not clearly defined in this manuscript. For example, the meaning of $LM_{2004\text{-}2009}$ in equation (1) is not presented. All variables and symbols should be explained in detail to avoid confusion.

**Response 3:** We apologize for not clearly defining this in the original manuscript. In the revised version, we will ensure that all equations are clearly explained, including the meaning of $LM_{2004\text{-}2009}$ in Equation (1), and that all variables and symbols are explicitly defined to avoid any confusion. Indeed $LM_{2004\text{-}2009}$ stands for Long-term Mean (LM) of total water storage from 2004 to 2009. Following details will be mentioned in the manuscript.

**3.1.2 Validation of GRACE-GWSC with Ref-GWSC**

GRACE TWSA are in centimetres (cm) of EWH which are converted to meters (m) of EWH. Thereafter, contemporaneous datasets of all variables i.e., quality controlled well measurements, storage anomalies from conopy, river, lake, reservoir, soil moisture, wet land, and snow, and GRACE TWSA are analysed. GRACE provides TWSA with respect to the mean TWS over 2004 to 2009, whereas other variables are not anomalies hence they are converted to anomalies by removing a Long-term Mean over 2004 to 2009 ($LM_{2004-2009}$). This can be implemented for all variables except well observations which are available only at seasonal interval. Therefore, to resolve this issue, we compute the differences between consecutive observations where CGWB GWL data is available, as the change in anomalies is equivalent to the change in the actual values at the corresponding times. This method provides a more robust and consistent basis for validation compared to interpolating the CGWB seasonal GWL data, which may introduce additional uncertainties and potential errors. The concept can be expressed mathematically as follows:

$$TWSA_i = TWS_i - LM_{2004-2009} \tag{3}$$
$$TWSC_{ij} = TWSA_i - TWSA_j = TWS_i - TWS_j \tag{4}$$
$$GRACE\ GWSC_{ij} = TWSC_{ij} - CSC_{ij} - SnSC_{ij} - SMSC_{ij} - SWSC_{ij} \tag{5}$$

where $i$ = present month (January/May/August/November) and $j$ represents the data for the previous month. Eq. 3 is used to compute TWSA for $i^{th}$ month ($TWSA_i$) where $TWS_i$ represent the TWS of $i^{th}$ the month. Eq.4 shows that $TWSC_{ij}$ i.e., TWS change (TWSC) obtained after subtracting TWSA of $j^{th}$ month ($TWSA_j$) from the $i^{th}$ month ($TWSA_i$) is equal to subtracting TWS of $j^{th}$ month ($TWS_j$) from the $i^{th}$ month ($TWS_i$) since the $LM_{2004-2009}$ cancels out. Using Eq. 5 GRACE-GWSC of $j^{th}$ month w.r.t. $i^{th}$ month ($GRACE\ GWSC_{ij}$) is computed after removal of Storage Changes due to Canopy ($CSC_{ij}$), Snow ($SnSC_{ij}$), Soil Moisture ($SMSC_{ij}$), and Surface Water Storage Changes ($SWSC_{ij}$) which includes storages of river streams, lakes, reservoirs, and wetlands. The resulting $GRACE\ GWSC_{ij}$ are validated against Ref-GWSC. To validate the GRACE product at its native resolution, median of quality controlled GWLs is computed over a mascon whereas to validate the downscaled product, median computation is done over a grid of 0.5° × 0.5°.

Further the downscaling methodology will be explained in a detailed manner in the revised manuscript as follows:

**3.2 Statistical downscaling to generate Mascon-wise Mass Conservation (MMC) downscaled product**

We used Partial Least square Regression (PLR) method to downscale GRACE TWSA to 0.5° × 0.5° from its native resolution of about 3° × 3°. Flow chart of the methodology is as shown in Figure 3. Our method is a modified version of the method elaborated in Vishwakarma et al. (2021) which uses regression model as shown in Eq.6.

$$S = LH \tag{6}$$

Where $S$ represent predictand matrix with $n$ (total number of months) × $g$ (total number of 0.5° × 0.5° grids in a mascon) dimensions. WGHM TWSA performs the role of this predictand matrix ($S$). $L$ is a predictor matrix i.e., observation matrix. Its dimensions are $n \times d$. Where $d$ represent columns of P, ET, R, and GRACE TWSA. Eq. 6 is solved to obtain $H$ (prediction matrix, $d \times g$). The modification we implemented here is that mass is conserved at mascon scale instead of catchment scale i.e., for whole set of operations Vishwakarma et al. (2021) unit was a catchment; in this study it is changed to a mascon.

To set up the predictor matrix ($L$) is the first and very important step here. $L$ consists of residuals of four variables (P, ET, R, and GRACE TWSA). In fact, there exist a temporal lead or lag amongst the water budget components (P, ET and R). Hence, K (=12) shifted time series of these datasets are generated and cyclostationary mean is removed to obtain residuals. To generate residuals of GRACE TWSA trend and annual cycle is removed from it.

Consider a mascon where total 36 0.5° × 0.5° grids are present. Our study period (January 2004 –December 2023) contains total 240 months. Thus, data matrix dimensions for a given mascon are 240 (rows) × 36 (columns). To generate 12 months shifted time series of P, ET and R, data from January 2003 is required. When $K = 1$, rows run from Jan 2003 to Dec 2022. For $K = 2$, row starts with Feb 2003 and ends at Jan 2023 and it goes on. To remove cyclostationary mean, month wise average is computed for the period between 2004 and 2023 resulting into 12 values which are removed from individual datasets. Thus, for a single variable after combining all 12 shifted timeseries, matrix dimensions become 240 × 432. Next, GRACE TWSA are detrended and annual cycle is removed to obtain residuals (240 × 36). Combining P, ET, R and GRACE TWSA generates our final observation matrix ($L$) with dimensions (240 × (1332=432+432+432+36)). Thereafter, residuals of WGHM TWSA are obtained after detrending and removal of GRACE annual cycle. This represents $S$ (240 × 36) in Eq. 6. Eq. 6 represents ideal case but in reality, measurements contain inherent noise. Thus, multivariate model becomes:

$$S = LH + E \tag{7}$$

Where $E$ in Eq. 7 represents the noise. Once Eq. 7 is set up, PLR obtains Principal Components (PCs) via Singular Value Decomposition (SVD). i.e., $H$ will be computed. Please note dimensions of $H$ is $d \times g$, which suggests grid wise coefficient will be determined unlike traditional regression method.

To solve Eq. 7, covariance matrix ($C$) is computed using Eq. (8) which is decomposed via SVD (Eq. 9).

$$C = L^T S \tag{8}$$

$$C = U_C \Sigma_C V_C^T \tag{9}$$

Where $U_C$ $(d \times r)$ and $V_C$ $(r \times d)$ are canonical modes (joint normalised eigen vectors for $L$ and $S$), $\Sigma_C$ $(r \times r)$ is a diagonal matrix which will contain covariance between $L$ and $S$. $r$ represents canonical modes from SVD. To obtain PCs of $L$ $(U_L)$ which are significantly correlated with $S$, $L$ is projected on $U_C$ using Eq. 10.

$$U_L = LU_C \tag{10}$$

Rearranging Eq. 10, Eq. 11 can be obtained as follows,

$$L = U_L U_C^T \tag{11}$$

Eq. 11 substituted to Eq.7 gives,

$$S = U_L U_C^T H + E \tag{12}$$

This can be written as,

$$S = U_L K + E \tag{13}$$

Where $K$ $(r \times g)$ is transformed regression matrix obtained by projecting $H$ on $U_C$. Now, we want to conserve the mass over a mascon unlike a catchment (Vishwakarma et al., 2021) even after downscaling, a mass budget constraint is applied here. i.e., $S = \Delta M$ = Average TWSA over a mascon.

$$\Delta M = U_L K + E \tag{14}$$

Finally bringing the constraint in the observation space and solving for $K$ using least squares method,

$$K = (X^T X)^{-1} X^T Y \tag{15}$$

$$X = \begin{pmatrix} U_L \\ U_L \end{pmatrix} \text{ and } Y = \begin{pmatrix} S \\ \Delta M \end{pmatrix} \tag{16}$$

In a simplified manner Eqs. 15,16 suggests that solutions should converge to $S$ and further only those solutions should be retained which will conserve the mass over a mascon. The $K$ obtained after solving Eq. 15 is then substituted in Eq.17 (obtained after rearranging $K = U_C^T H$ ) to get $H$.

$$H = U_c K \tag{17}$$

This computed $H$ when put back to Eq. 6 with already prepared $L$ matrix, downscaled GRACE residuals are obtained. To which trend and annual cycle are added back to get final downscaled GRACE TWSA referred as MMC downscaled product.

Section 3.3 will be modified to justify the implementation of evaluation metric as follows:

**3.3 Evaluation of downscaled products**

First, validation of all three downscaled products is performed using Ref-GWSC measurements by computing the Correlation Coefficient (r) and Root-Mean-Square Error (RMSE). In addition to these classical evaluation metrics, the temporal metric gain proposed by Pascal et al. (2022) is implemented to compare the performance of the downscaled products. As noted by Pascal et al. (2022) most studies do not assess the performance of downscaled GRACE products relative to the original GRACE data. By using the temporal gain metric, we can quantitatively evaluate the accuracy of the downscaled products.

In the present study, GRACE JPL mascon data are used as the low-resolution reference without applying the scale factor. These scale factors, being multiplicative and model-based, can amplify any uncertainties in the GRACE product if the model does not align with reality. Vishwakarma et al. (2017) and Pascal et al. (2022) have shown that applying the scale factor can degrade rather than improve the results. Consequently, we have not extended the validation framework to compute spatial gain, since our low-resolution reference does not incorporate the scale factor and thus does not contain spatial variability. As illustrated in Figure, the reference values are uniform over each mascon.

[Figure]

**Figure.** GRACE mascons over India for September 2004. Red boxes represent mascon boundaries.

**Comment 4:** [Results and Discussion] The performance varies significantly across mascons, with some showing improvement and others even negative correlation. These spatial differences are only described in this manuscript, but they are not explained in sufficient detail. Potential reasons, such as groundwater abstraction, geological conditions, or climatic variability, should be discussed in this part.

**Response 4:** We thank the reviewer for this comment. In the revised manuscript, we will discuss the reasons for the varying performance across mascons, including factors such as groundwater abstraction, geological conditions, and climate variability, to provide a clearer interpretation of the results.

For example, Section 4.1 will be updated as shown below.

**4.1 Validation statistics for GRACE GWSC at native resolution ($3° \times 3°$) across India**

Figure indicates mascon-wise maps of r and RMSE between GRACE GWSC and ref-GWSC across India. The validation is performed only for 36 mascons out of 52 owing to missing well observations. Out of 36, 28 mascons showed positive r ranging from 0.06 to 0.91. Mascons 5 and 27 showed a maximum r value of 0.91, indicating that GRACE is capable of explaining 83% of GWSC variability in these mascons. Mascons 19, 26, 35, 36 and 41 showed r values varying between 0.83 to 0.86 explaining the variability of GWSC between 69% to 74%. For mascons 9, 10, 15, 18, 24, 25, 28, 34, 37, 40, 45, 49 r is ranging between 0.63 and 0.79, explaining 40 to 60% of GWSC variability. These results indicates that GRACE GWSC is capable of explaining variability in GWSC over 40% for majority of the mascons. In contrast, mascons 16, 32, 43, and 48 showed low r values (0.15, 0.06, 0.17 and 0.10, respectively), indicating that GRACE is not even capturing a 3% of GWSC variability. Mascons 4, 8, 17, 22, 23, 44 and 50 exhibited negative correlations. RMSE values observed to be ranging between 0.06 m to 0.32 m. The best (minimum) RMSE is observed for mascon 6 whereas worst (maximum) is shown by mascon 16. Overall RMSE remained low for majority of the mascons (< 0.16 m) whereas mascon 9, 16, 17 showed higher RMSE varying between 0.29 m and 0.32 m.

Interestingly, we also found that the validation performance is independent of number and distribution of wells in the mascon. For instance, mascon 17, with 112 wells, exhibited a negative correlation of -0.32, whereas mascon 5, with only 33 wells showed maximum correlation of 0.91. Similarly, mascon 15, where all wells are clutered in the right side of the mascon, still showed a good correlation of 0.63. Whereas mascon 16 has a fairly well distributed network of wells, still showed a poor correlation of 0.15. Thus, the failure of GRACE in capturing groundwater signal in some mascons can be attributed to multiple factors such as, errors in measurements of GWL, high irrigational activities occurring at local scales, inadequacy of WGHM model in computing exact storages of canopy, snow, soil moisture, river streams, lakes, reservoirs, and wetlands. Conversely, in some mascons GRACE performs exceptionally well, achieving r = 0.91 and RMSE = 0.09 m. The strong agreement also underscores the superiority of the difference-based validation method, as it preserves the integrity of observed

data and yields more reliable results than approaches that rely on temporal interpolation of CGWB measurements.

[Figure]

**Figure.** Mascon-wise maps of (a) the correlation coefficient (r) and (b) RMSE between GRACE GWSC and Ref-GWSC over India.

**Comment 5:** [Results and Discussion] The MMC method fails for mascon 9, but no further explanation is given in this part. A discussion of why the method fails here would be valuable.

**Response 5:** Thanks for the comment. This was a misunderstanding as the color scale was limited between -0.2 to 0.2 m. Here is the figure with modified extended scale where spatial variability in mascon 9 is clearly visible.

[Figure]

**Figure.** Mascon-wise Mass Conservation (MMC) downscaled TWSA (m) for September 2004.

*Please note that additionally, in the revised manuscript, we will extend the Mascon-wise Mass Conservation (MMC) downscaled product up to December 2023, thereby improving the temporal coverage of the dataset.*

**Comment 6:** [Results and Discussion] The manuscript does not adequately discuss uncertainties or limitations of the proposed method, including sparse well distribution, specific yield estimation, errors in the forcing data, and model bias.

**Response 6:** We appreciate the reviewer's comment. In the revised manuscript, we will discuss the main uncertainties and limitations of our method.

For example, Secti. 4.1 will be updated as shown below. Similarly, the downscaling results and their associated uncertainties will be presented and discussed in detail in the revised manuscript.

**4.1 Validation statistics for GRACE GWSC at native resolution (3° × 3°) across India**

Figure indicates mascon-wise maps of r and RMSE between GRACE GWSC and ref-GWSC across India. The validation is performed only for 36 mascons out of 52 owing to missing well observations. Out of 36, 28 mascons showed positive r ranging from 0.06 to 0.91. Mascons 5 and 27 showed a maximum r value of 0.91, indicating that GRACE is capable of explaining 83% of GWSC variability in these mascons. Mascons 19, 26, 35, 36 and 41 showed r values varying between 0.83 to 0.86 explaining the variability of GWSC between 69% to 74%. For mascons 9, 10, 15, 18, 24, 25, 28, 34, 37, 40, 45, 49 r is ranging between 0.63 and 0.79, explaining 40 to 60% of GWSC variability. These results indicates that GRACE GWSC is capable of explaining variability in GWSC over 40% for majority of the mascons. In contrast, mascons 16, 32, 43, and 48 showed low r values (0.15, 0.06, 0.17 and 0.10, respectively), indicating that GRACE is not even capturing a 3% of GWSC variability. Mascons 4, 8, 17, 22, 23, 44 and 50 exhibited negative correlations. RMSE values observed to be ranging between 0.06 m to 0.32 m. The best (minimum) RMSE is observed for mascon 6 whereas worst (maximum) is shown by mascon 16. Overall RMSE remained low for majority of the mascons (< 0.16 m) whereas mascon 9, 16, 17 showed higher RMSE varying between 0.29 m and 0.32 m.

Interestingly, we also found that the validation performance is independent of number and distribution of wells in the mascon. For instance, mascon 17, with 112 wells, exhibited a negative correlation of -0.32, whereas mascon 5, with only 33 wells showed maximum correlation of 0.91. Similarly, mascon 15, where all wells are clutered in the right side of the mascon, still showed a good correlation of 0.63. Whereas mascon 16 has a fairly well distributed network of wells, still showed a poor correlation of 0.15. Thus, the failure of GRACE in capturing groundwater signal in some mascons can be attributed to multiple factors such as, errors in measurements of GWL, high irrigational activities occurring at local scales, inadequacy of WGHM model in computing exact storages of canopy, snow, soil moisture, river streams, lakes, reservoirs, and wetlands. Conversely, in some mascons GRACE performs exceptionally well, achieving r = 0.91 and RMSE = 0.09 m. The strong agreement also underscores the superiority of the difference-based validation method, as it preserves the integrity of observed data and yields more reliable results than approaches that rely on temporal interpolation of CGWB measurements.

[Figure]

**Figure.** Mascon-wise maps of (a) the correlation coefficient (r) and (b) RMSE between GRACE GWSC and Ref-GWSC over India.

**Comment 7:** [Results and Discussion] The conclusions are too general. The practical implications for groundwater management and policy are not sufficiently discussed and should be strengthened.

**Response 7:** In the revised manuscript, we will strengthen the conclusions by highlighting the practical implications of our findings for groundwater management and policy, emphasizing how the downscaled product can inform regional water resource planning and decision-making. In the revised manuscript, the following modified conclusion will be inserted:

**5 Conclusions**

In this study, we carried out validation and downscaling of GRACE observations over India. For proper validation of GWSC specifically contributions from other storage components such as canopy, snow, soil moisture, river streams, lakes, reservoirs, and wetlands removed from GRACE TWSA. We conclude that GRACE at its native resolution (about 3° × 3°) when compared with in-situ well data showed varied performance over mascons. At some mascons GRACE successfully captures the observed variations with high r (> 0.86) and RMSE (< 0.16 m) values. However, there were a few mascons where GRACE didn't perform well possibly due to errors in GWL data and inadequate accounting of water storages other than groundwater. By implementing modified downscaling approach, a new downscaled product (MMC) is developed and compared with CMC. Over several mascons MMC approach showed a drastic improvement by converting negative correlation seen in CMC into positive ones over CMC approach. The DL approach also performs better than CMC but in terms of RMSE, MMC outperformed both CMC and DL when validated with in-situ.

The resulting high-resolution groundwater storage datasets hold immense potential for a wide range of scientific and practical applications in water resource research and management. By providing spatially varied information on groundwater, they can significantly improve drought characterization and severity mapping, especially in regions with sparse in-situ data. These downscaled products can serve as critical inputs to hydrologic and groundwater models, enabling more accurate representation of subsurface storage dynamics at regional and basin scales. These all together can help in evaluating effectiveness of groundwater recharge/extraction policies. The fine resolution information can further support in basin-planning, watershed restorations and river rejuvenation initiatives by helping to identify zones of water stress and potential recharge areas. Consequently,

the developed downscaled storage product can play a crucial role in guiding sustainable groundwater management, enhancing climate resilience and informed evidence-based decision making for India's ongoing water resource and rejuvenation program.

**Minor Comments**

**Comment 8**: [Introduction, Line 22 and Line 26] Several language errors exist in this manuscript (e.g., "Gao and Soja" should be "Gou and Soja"). Careful proofreading is required.

**Response 8:** We thank the reviewer for pointing this out. In the revised manuscript, we will carefully proofread the manuscript and correct all language errors, including reference names such as "Gao and Soja," which will be corrected to "Gou and Soja."

**Comment 9:** [Results and Discussion, Figure 6] The presentation of "0.5°x0.5°" is questionable and should be replaced with the multiplication sign (×).

**Response 9:** Thanks. In the revised manuscript, we will correct the presentation of "0.5°x0.5°" and replace it with the proper multiplication sign (×) for clarity and accuracy. The same will be followed throughout the manuscript:

**Comment 10:** [Results and Discussion] Too many boxplots are included in this manuscript. Consider integrating these figures to make the results more clear.

**Response 10:** We thank the reviewer for this suggestion. In the revised manuscript, we will introduce following heatmaps replacing box plots for more clarity. We have replotted them and are shown below for your reference.

[Figure]

**Figure.** Left panel showing Correlation Coefficient (r) for GRACE, CMC and MMC. Right panel showing Root Mean Square Error (RMSE) for GRACE, CMC and MMC.

Similar heat maps will be generated for displaying gain in r and RMSE for all three products (GRACE, CMC and MMC). Also, to show r between products (CMC, MMC) and WGHM heat maps will be generated. However, to display DL results we will use the box plot just to break the monotony.

**Comment 11:** [References] The reference list is inconsistent in format. Please carefully check the references throughout the manuscript.

**Response 11:** We apologize for the inconsistencies in the reference list. In the revised manuscript, we will carefully check and standardize all references to ensure consistent formatting throughout.

**References cited in this response:**

Ali, S., Liu, D., Fu, Q., Cheema, M. J. M., Pham, Q. B., Rahaman, M. M., Dang, T. D., and Anh, D. T.: Improving the Resolution of GRACE Data for Spatio-Temporal Groundwater Storage Assessment, Remote Sens., 13, 3513, https://doi.org/10.3390/rs13173513, 2021.

Arshad, A., Shafeeque, M., Tran, T. N. D., Mirchi, A., Xiang, Z., He, C., AghaKouchak, A., Besnier, J., and Rahman, M. M.: Multi-model ensemble machine learning-based downscaling and projection of GRACE data reveals groundwater decline in Saudi Arabia throughout the 21st century, J. Hydrol.: Reg. Stud., 60, 102552, https://doi.org/10.1016/j.ejrh.2025.102552, 2025.

Bhanja, S. N., Mukherjee, A., Saha, D., Velicogna, I., and Famiglietti, J., S.: Validation of GRACE based groundwater storage anomaly using in-situ groundwater level measurements in India, J. Hydrol., 543, 729-738, https://doi.org/10.1016/j.jhydrol.2016.10.042, 2016.

Chen, L., He, Q., Liu, K., Li, J., and Jing, C.: Downscaling of GRACE-Derived Groundwater Storage Based on the Random Forest Model, Remote Sens., 11, 2979, https://doi.org/10.3390/rs11242979, 2019.

Gemitzi, A., Koutsias, N., and Lakshmi, V.: A spatial downscaling methodology for GRACE total water storage anomalies using GPM IMERG precipitation estimates, Remote Sens., 13, 5149, https://doi.org/10.3390/rs13245149, 2021.

Gorugantula, S. S., and Kambhammettu, B. P.: Sequential downscaling of GRACE products to map groundwater level changes in Krishna River basin, Hydrolog. Sci. J., 67, 1846-1859, https://doi.org/10.1080/02626667.2022.2106142, 2022.

Gou, J., and Soja, B.: Global high-resolution total water storage anomalies from self-supervised data assimilation using deep learning algorithms, Nature Water ,2, 139-150,https://doi.org/10.1038/s44221-024-00194-w, 2024.

Jyolsna, P. J., Kambhammettu, B. V. N. P., and Gorugantula, S.: Application of random forest and multi-linear regression methods in downscaling GRACE derived groundwater storage changes, Hydrolog. Sci. J., 66, 874–887, https://doi.org/10.1080/02626667.2021.1896719, 2021.

Kalu, I., Ndehedehe, C. E., Ferreira, V. G., Janardhanan, S., Currell, M. and Kennard, M. J.: Statistical downscaling of GRACE terrestrial water storage changes based on the Australian Water Outlook model. Sci. Rep., 14, 10113, https://www.nature.com/articles/s41598-024-60366-2, 2024.

Karunakalage, A., Sarkar, T., Kannaujiya, S., Chauhan, P., Pranjal, P., Taloor, A. K., and Kumar, S.: The appraisal of groundwater storage dwindling effect, by applying high resolution downscaling GRACE data in and around Mehsana district, Gujarat, India, Groundwater Sustain. Dev., 13, 100559, https://doi.org/10.1016/j.gsd.2021.100559, 2021.

Kuruva, S. K., Suryawanshi, M. R., Shakya, A., Va, C., Shaw, B., Sukumaran, V., Kour, R., Kochar, A., Chander, S., Nikam, B. R., and Dasika, N. K.: Quality controlled, reliable groundwater level data with corresponding specific yield over India, Scientific Data, 12, 1609, https://doi.org/10.1038/s41597-025-05899-5, 2025.

Miro, M. E., and Famiglietti, J. S.: Downscaling GRACE remote sensing datasets to high-resolution groundwater storage change maps of California's Central Valley. Remote Sens., 10, 1-18, https://doi.org/10.3390/rs10010143, 2018.

Ning, S., Ishidaira, H., and Wang, J.: Statistical Downscaling of Grace-Derived Terrestrial Water Storage Using Satellite and Gldas Products, J. Jpn. Soc. Civ. Eng. Ser. B1, 70, I_133–I_138, https://doi.org/10.2208/jscejhe.70.I_133, 2014.

Pascal, C., Ferrant, S., Selles, A., Maréchal, J.-C., Paswan, A., and Merlin, O.: Evaluating downscaling methods of GRACE (Gravity Recovery and Climate Experiment) data: a case study over a fractured crystalline aquifer in southern India, Hydrol. Earth Syst. Sci., 26, 4169–4186, https://doi.org/10.5194/hess-26-4169-2022, 2022.

Sarkar, T., Kannaujiya, S., Taloor, A. K., Ray, P. K. C. and Chauhan, P.: Integrated study of GRACE data derived interannual groundwater storage variability over water stressed Indian regions. Groundwater Sustain. Dev., 10, 100376, https://doi.org/10.1016/j.gsd.2020.100376, 2020.

Scanlon, B. R., Zhang, Z., Save, H., Wiese, D. N., Landerer, F. W., Long, D., Longuevergne, L., and Chen, J.: Global evaluation of new GRACE mascon products for hydrologic applications, Water Resour. Res., 52, 9412-9429, https://doi.org/10.1002/2016WR019494, 2016.

Schumacher, M., Forootan, E., van Dijk, A. I. J. M., Müller Schmied, H., Crosbie, R. S., Kusche, J., and Döll, P.: Improving drought simulations within the Murray-Darling Basin by combined calibration/assimilation of GRACE data into the WaterGAP Global Hydrology Model, Remote Sens. Environ., 204, 212–228, https://doi.org/10.1016/j.rse.2017.10.029, 2018.

Sun, J., Hu, L., Cao, X., Liu, D., Liu, X. and Sun, K.: A dynamical downscaling method of groundwater storage changes using GRACE data, J. Hydrol.: Reg. Stud, 50, 101558, https://doi.org/10.1016/j.ejrh.2023.101558, 2023.

Vishwakarma, B. D., Horwath, M., Devaraju, B., Groh, A., and Sneeuw, N.: A data-driven approach for repairing the hydrological catchment signal damage due to filtering of GRACE products, Wat. Resour. Res., 53, 9824-9844, https://doi.org/10.1002/2017WR021150, 2017.

Vishwakarma, B. D., Zhang, J., and Sneeuw, N.: Downscaling GRACE total water storage change using partial least squares regression, Scient. Data, 8, 95, https://doi.org/10.1038/s41597-021-00862-6, 2021.

Yin, W., Hu, L., Zhang, M., Wang, J., and Han, S.-C.: Statistical Downscaling of GRACE-Derived Groundwater Storage Using ET Data in the North China Plain, J. Geophys. Res.-Atmos., 123, 5973–5987, https://doi.org/10.1029/2017JD027468, 2018.